# Uncovering the Spectrum of Graph Generative Models: From One-Shot to Sequential

## Abstract

In the field of deep graph generative models, two families coexist: one-shot models, which fill the graph content in one go given a number of nodes, and sequential models, where new nodes and edges are inserted sequentially and autoregressively. Recently, one-shot models are seeing great popularity due to their rising sample quality and lower sampling time compared to the more costly autoregressive models. With this paper we unify the two worlds in a single framework, unlocking the whole spectrum of options where one-shot and sequential models are but the two extremes. We use the denoising diffusion models' theory to develop a node removal process, which destroys a given graph through many steps. An insertion model reverses this process by predicting how many nodes have been removed from the intermediate subgraphs. Then, generation happens by iteratively adding new blocks of nodes, with size sampled from the insertion model, and content generated using any one-shot model. By adjusting the knob on node removal, the framework allows for any degree of sequentiality, from one-shot to fully sequential, and any node ordering, e.g., random and BFS. Based on this, we conduct the first analysis of the sample quality-time trade-off across a range of molecular and generic graphs datasets. As a case study, we adapt DiGress, a diffusion-based one-shot model, to the whole spectrum of sequentiality, reaching new state of the art results, and motivating a renewed interest in developing autoregressive graph generative models.

## 1 Introduction

Graphs are mathematical objects that allow us to represent any kind of structured data, where components and their relationships can be identified. They are used in many domains: social networks, chemical structures, visual scenes, and to represent knowledge. For this reason, there was always motivation to study how to generate new graphs following domain-specific rules. This was the case for Erdos Rényi random graphs (Erdős et al., 1960), and now with Deep Graph Generative Models.

In this field, two families of models can be identified: (1) one-shot models, generating the entire adjacency matrix and node features of a graph in one go; (2) sequential models, generating nodes and edges one after the other. The literature of one-shot models is very rich, with the graph equivalents of Variational Autoencoders (Simonovsky & Komodakis, 2018), Normalizing Flows (Zang & Wang, 2020), and Diffusion Models (Ho et al., 2020). In parallel, the same techniques have been applied in sequential models (Liao et al., 2019; Luo et al., 2021; Kong et al., 2023), just with an additional assumption: graphs are sequences of nodes to generate, and each one depends on the nodes before. This autoregressive property may spark the thought that a better sample quality can be achieved. Looking into the literature, this is not always the case, as one-shot models have entered the state-of-the-art on challenging datasets like ZINC250k (Irwin et al., 2012), Ego (Sen et al., 2008), and many more. However, one-shot models need to sample the number of nodes to generate from a histogram pre-computed from the dataset, which can not be ideal in a conditional setup. In this sense, autoregressive sequential models are more flexible in that they also learn the size distribution.

To bridge the gap between these two seemingly different modalities of generation, we propose a new diffusion-based framework called Insert-Fill-Halt (IFH, Figure 1), which reveals the whole spectrum of *sequentiality*. At each step, the Insertion Model chooses how many new nodes to generate, the Filler Model how to fill the new nodes' labels, features, and connections, and the Halt Model chooses

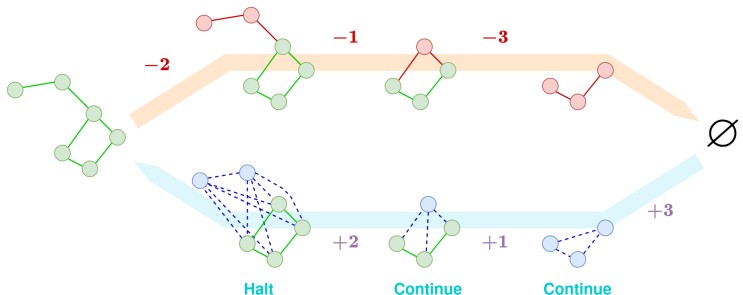

Figure 1: Our Insert-Fill-Halt model. During training, a graph is corrupted (left to right) by iteratively removing nodes until the empty graph is left. At each step, the insertion (violet), filler (blue), and halt (cyan) models have to predict how many nodes were removed, what content they had, and whether the graph is terminal for generation.

if generation needs to terminate. The three components can be trained by applying a node removal process to a graph data point, which the models try to undo. It can be shown that one-shot models are 1-step IFH models, and that any degree of sequentiality can be achieved by choosing different removal processes, down to 1 node at a time. The framework also allows to adapt one-shot models to act as Filler Models, with the hope of inheriting or even improving their sample quality.

We show empirically that adapting Digress (Vignac et al., 2022), a state-of-the-art graph generative model, to 1-node sequential, leads to surpassing all autoregressive baselines, and is competitive with other state-of-the-art one-shot baselines such as CDGS (Huang et al., 2023). Other than providing a way of unifying the two theories, we also want to spur a new interest in autoregressive graph generative models, which might actually be the better choice for conditional generation.

## 2  BACKGROUND AND RELATED WORK

Let $\mathcal{G} = (\mathcal{V}, \mathcal{E})$ be a graph, where $\mathcal{V} = \{v_1, v_2, \ldots, v_n\}$ is the set of vertices, and $\mathcal{E} = \{(v_i, v_j) \mid v_i, v_j \in \mathcal{V}, \ v_i, v_j \text{ are linked}\}$ is the set of edges. The number of nodes and edges of $\mathcal{G}$ are respectively denoted $n = |\mathcal{V}|$ and $m = |\mathcal{E}|$. An alternative representation for $\mathcal{E}$ is the adjacency matrix $\boldsymbol{A} \in \{0, 1\}^{n \times n}$ where $A_{i,j} = 1$ if $(v_i, v_j) \in \mathcal{E}$, and 0 otherwise. In the case of undirected graphs, edges are represented not by tuples but by sets $\{v_i, v_j\}$, and $\boldsymbol{A}$ is a symmetric matrix, i.e., $\boldsymbol{A} = \boldsymbol{A}^\top$. In labeled graphs, $\mathcal{V}$ and $\mathcal{E}$ are coupled with node features $\boldsymbol{X} \in \mathbb{R}^{n \times d_x}$ and edge features $\mathsf{E} \in \mathbb{R}^{n \times n \times d_e}$, where $d_x$ and $d_e$ are the dimensions of a single node/edge feature vector respectively. Global features $\boldsymbol{y} \in \mathbb{R}^{d_y}$ of the graph can also be included.

**Definition 1** (Remove operation). *Removing a node $v_i$ from $\mathcal{G}$ is equivalent to removing $v_i$ from $\mathcal{V}$, its entry in $\boldsymbol{X}$, all edges $(v_i, v_j)$ or $(v_j, v_i)$ from $\mathcal{E}$ in which $v_i$ participates, and the row and column in $\mathsf{E}$ assigned to its connectivity.*

**Definition 2** (Induced subgraph). *A subgraph $\mathcal{G}_A$ induced in $\mathcal{G}$ by $\mathcal{V}_A \subseteq \mathcal{V}$ is the subgraph obtained by removing all nodes in $\mathcal{V}_B = \mathcal{V} \setminus \mathcal{V}_A$ from $\mathcal{G}$.*

**Definition 3** (Split operation). *A split $(\mathcal{G}_A, \mathcal{G}_B, \mathcal{E}_{AB}, \mathcal{E}_{BA})$ of $\mathcal{G}$ through $\mathcal{V}_A$ is the tuple composed by the subgraphs $\mathcal{G}_A, \mathcal{G}_B$ induced by $\mathcal{V}_A$ and $\mathcal{V}_B = \mathcal{V} \setminus \mathcal{V}_A$, the intermediate edges $\mathcal{E}_{AB}$ linking nodes in $\mathcal{V}_A$ to nodes in $\mathcal{V}_B$ and vice versa for $\mathcal{E}_{BA}$.*

**Definition 4** (Merge operation). *Given a tuple $(\mathcal{G}_A, \mathcal{G}_B, \mathcal{E}_{AB}, \mathcal{E}_{BA})$, the merged graph $\mathcal{G}$ is defined with $\mathcal{V} = \mathcal{V}_A \cup \mathcal{V}_B$ and $\mathcal{E} = \mathcal{E}_A \cup \mathcal{E}_B \cup \mathcal{E}_{AB} \cup \mathcal{E}_{BA}$. Node and edge features are concatenated as shown in Figure 2.*

Splitting implies a separation also on features: $\boldsymbol{X}_A$ and $\boldsymbol{X}_B$ for nodes, $\mathsf{E}_{AA}, \mathsf{E}_{AB}, \mathsf{E}_{BA}$, and $\mathsf{E}_{BB}$ for edges, as shown in Figure 2. When splitting undirected graphs, it immediately follows that $\mathcal{E}_{AB} = \mathcal{E}_{BA}$ and $\mathsf{E}_{AB} = \mathsf{E}_{BA}^\top$. A merge operation reverses a split operation: in that case, node and edge features are concatenated as shown in Figure 2. Now we can define the main object for our mathematical framework, the forward and reversed removal sequences.

**Definition 5** (Forward and reversed removal sequence). *A graph sequence $\mathcal{G}_{0:T}^{\rightarrow} = (\mathcal{G}_t)_{t=0}^T$ is a forward removal sequence of $\mathcal{G}$ when $\mathcal{G}_0 = \mathcal{G}$, $\mathcal{G}_T$ is the empty graph $\varnothing$, and $\mathcal{G}_t$ is an induced*

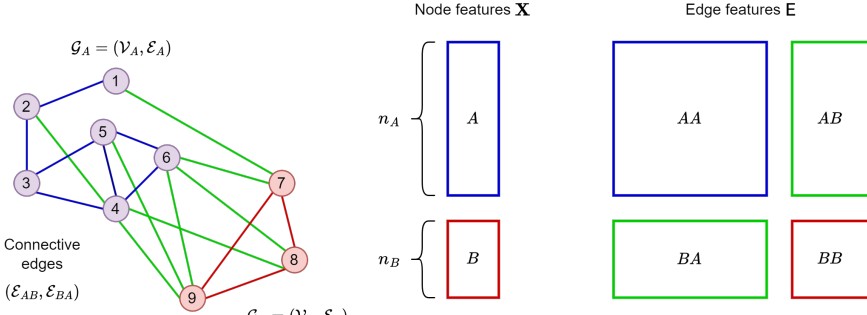

Figure 2: Split operation. In blue and red are the induced subgraphs $\mathcal{G}_A$ and $\mathcal{G}_B$. In green are the intermediate edges $\mathcal{E}_{AB}, \mathcal{E}_{BA}$. On the right is the split adjacency matrix, with the same coloring.

subgraph of $\mathcal{G}_{t-1}$ for all $t = 1, \ldots, T$. $\mathcal{G}_{0:T}^{\leftarrow}$ is a reversed removal sequence of $\mathcal{G}$ if it is a sequence $\mathcal{G}_{0:T}^{\rightarrow}$ of $\mathcal{G}$ navigated in reverse, i.e., with index $s = T - t$. In this case $\mathcal{G}_{s-1}$ is an induced subgraph of $\mathcal{G}_s$ for all $s = 1, \ldots, T$.

We denote $\mathcal{F}(\mathcal{G}, T)$ and $\mathcal{R}(\mathcal{G}, T)$ as the sets of all forward and reversed removal sequences of $\mathcal{G}$ of length $T$. For the halting processes we borrow the notation from Banino et al. (2021).

**Definition 6** (Halting process). *A halting process $\Lambda_s$ is a Markov process where, at each time step, $\Lambda_s$ is a Bernoulli random variable with outcomes $0, 1$ (continue, halt), and evolves as follows: it starts with $\Lambda_0 = 0$ (continue), and proceeds with Markov transitions $p(\Lambda_s = 1 | \Lambda_{s-1} = 0) = \lambda(s)$ until step $s = T$ the process is absorbed in state 1 (halt), i.e., $p(\Lambda_s = 1 | \Lambda_{s-1} = 1) = 1 \, \forall s > 0$.*

## 2.1 RELATED WORKS

Given a set of observable graph data points with unknown distribution $p_{\text{data}}(\mathcal{G})$, likelihood maximization methods aim to learn the parameters $\theta$ of a model $p_\theta(\mathcal{G})$ to approximate the true distribution $p_{\text{data}}(\mathcal{G})$. In the context of deep graph generation (Guo & Zhao, 2022), $p_\theta(\mathcal{G})$ has been modeled as: (1) a sequential process, with nodes being added one by one or block by block autoregressively; (2) a one-shot process, generating the whole matrix structure of the graph in parallel.

**Sequential models** Sequential models frame graph generation as forming a trajectory $\mathcal{G}_{0:T}^{\leftarrow} = (\mathcal{G}_0, \ldots, \mathcal{G}_s, \ldots, \mathcal{G}_T)$ of increasingly big graphs, where $\mathcal{G}_0$ is usually an empty graph, $\mathcal{G}_T$ is the generation result, and the transition from $\mathcal{G}_{s-1}$ to $\mathcal{G}_s$ introduces new nodes and edges, without touching what is already in $\mathcal{G}_{s-1}$. In the case of node-sequence generation, $T$ is exactly the number of nodes $n$ in $\mathcal{G}$, and the transition $p(\mathcal{G}_t | \mathcal{G}_{t-1})$ appends exactly one node and the edges from that node to $\mathcal{G}_{t-1}$. In motif-sequence generation, blocks of nodes are inserted, together with new rows of the adjacency matrix. For the remainder of the paper, we will denote as *sequential* the models based on a node-sequence generation, *block-sequential* for motif-based models, and *autoregressive models* for addressing both. Given a halting criteria $\lambda_\nu(\mathcal{G}_s, s)$ (as defined in 6) based on current graph $\mathcal{G}_s$, the model distribution for a sequential model is of the form:

$$p_{\theta,\nu}(\mathcal{G}) = \sum_{\mathcal{G}_{0:n}^{\leftarrow} \in \mathcal{R}(\mathcal{G},n)} \lambda_\nu(\mathcal{G}_n, n) p_\theta(\mathcal{G}_n | \mathcal{G}_{n-1}) \prod_{s=1}^{T-1} (1 - \lambda_\nu(\mathcal{G}_s, s)) p_\theta(\mathcal{G}_s | \mathcal{G}_{s-1}) \quad (1)$$

An important ingredient in training autoregressive models is the node ordering, i.e., assigning a permutation $\pi$ to the $n$ nodes in $\mathcal{G}$ in order to build the trajectory $\mathcal{G}_{0:T}^{\leftarrow}$. With random ordering, the model has to explore each of the $n!$ possible permutations, whereas canonical orderings such as Breadth First Search (BFS) (You et al., 2018), Depth First Search (DFS), and many others (Liao et al., 2019; Chen et al., 2021) decrease the size of the search space, empirically increasing sample quality, although introducing an inductive bias which might not fit every case.

In the sequential model family fall generative RNN models such as GraphRNN (You et al., 2018) and block-sequential GRAN (Liao et al., 2019), autoregressive normalizing flow models like GraphAF (Shi et al., 2020) and its discrete variant GraphDF (Luo et al., 2021). Regarding block generation,

the work by Liao et al. (2019) is a precursor of ours, as the authors investigated the use of different fixed block sizes in a domain-agnostic setup with grid graphs. Kong et al. (2023) framed the task of finding the optimal node ordering as learning an absorbing discrete diffusion process that masks nodes one at a time, coupled with a denoising network generating new nodes in the reverse order.

**One-shot models** One-shot models employ a decoder network that maps a latent vector $z$ to the resulting graph $\mathcal{G}$. The latent vector is usually sampled from a tractable distribution (such as a Normal distribution), and the number of nodes is either fixed, sampled from the frequencies of nodes in the dataset, or predicted from the latent code $z$. In general, one-shot models have the form:

$$p_{\theta,\phi}(\mathcal{G}) = p_\theta(\mathcal{G}|n)p_\phi(n). \tag{2}$$

When $p_\theta(\mathcal{G}|n)$ is implemented by a neural network architecture that is equivariant to permutations in the order of nodes, then no node orderings are needed.

For one-shot generation, the classic generative paradigms are applied: VAE with GraphVAE (Simonovsky & Komodakis, 2018), GAN with MolGAN (De Cao & Kipf, 2018), Normalizing Flows with MoFlow (Zang & Wang, 2020), diffusion with EDP-GNN (Niu et al., 2020), discrete diffusion with DiGress (Vignac et al., 2022), energy-based models with GraphEBM (Liu et al., 2021), Stochastic Differential Equations (SDE) with GDSS (Jo et al., 2022) and CDGS (Huang et al., 2023).

### 2.1.1 DIFFUSION MODELS

We briefly introduce the denoising diffusion model (Sohl-Dickstein et al., 2015; Ho et al., 2020), as its theory will serve as the foundation of our framework. Let $\mathbf{x}_0$ be a data point sampled from an unknown distribution $q(\mathbf{x}_0)$. Denoising diffusion models are latent variable models with two components: (1) a diffusion process gradually corrupts $\mathbf{x}_0$ in $T$ steps with Markov transitions $q(\mathbf{x}_t|\mathbf{x}_{t-1})$ until $\mathbf{x}_T$ has some simple, tractable distribution $p_\theta(\mathbf{x}_T)$ (e.g. a Normal distribution); (2) a learned reverse Markov process with transition $p_\theta(\mathbf{x}_{t-1}|\mathbf{x}_t)$ denoises $\mathbf{x}_T$ back to the original data distribution $q(\mathbf{x}_0)$. The trajectories formed by the two processes are:

$$\underbrace{q(\mathbf{x}_{1:T}|\mathbf{x}_0) = \prod_{t=1}^{T} q(\mathbf{x}_t|\mathbf{x}_{t-1})}_{\text{Forward process}}, \qquad \underbrace{p_\theta(\mathbf{x}_{0:T}) = p_\theta(\mathbf{x}_T)\prod_{t=1}^{T} p_\theta(\mathbf{x}_{t-1}|\mathbf{x}_t)}_{\text{Reverse process}} \tag{3}$$

For $T \to +\infty$, the forward and reverse transitions share the same functional form (Feller, 1949), and choosing $q(\mathbf{x}_T|\mathbf{x}_0) = q(\mathbf{x}_T)$ allows in fact to easily sample $\mathbf{x}_T$. The distribution $p_\theta(\mathbf{x}_0)$ can be made to fit the data distribution $q(\mathbf{x}_0)$ by minimizing the variational upper bound:

$$L_{\text{vub}} = \mathbb{E}_{\mathbf{x}_0 \sim q(\mathbf{x}_0)}\bigg[ \underbrace{D_{\text{KL}}\big(q(\mathbf{x}_T|\mathbf{x}_0)\|p(\mathbf{x}_T)\big)}_{L_T} + \sum_{t=2}^{T} \underbrace{D_{\text{KL}}\big(q(\mathbf{x}_{t-1}|\mathbf{x}_t,\mathbf{x}_0)\|p_\theta(\mathbf{x}_{t-1}|\mathbf{x}_t)\big)}_{L_{t-1}}$$
$$\underbrace{-\mathbb{E}_{\mathbf{x}_1 \sim q(\mathbf{x}_1|\mathbf{x}_0)}\big[\log p_\theta(\mathbf{x}_0|\mathbf{x}_1)\big]}_{L_0} \bigg]. \tag{4}$$

Two necessary properties to make diffusion models feasible are for $q(\mathbf{x}_t|\mathbf{x}_0)$ and $q(\mathbf{x}_{t-1}|\mathbf{x}_t,\mathbf{x}_0)$ to have a closed form formula, in order to respectively (1) efficiently sample many time steps in parallel and (2) compute the KL divergences.

The first successful attempt with diffusion models defined the transitions as $q(\mathbf{x}_t|\mathbf{x}_{t-1}) = \mathcal{N}(\mathbf{x}_t; \sqrt{1-\beta_t}\mathbf{x}_{t-1}, \beta_t\boldsymbol{I})$ (Ho et al., 2020) where $\beta_t$ is a variance schedule. Later, diffusion models were adapted for discrete spaces (Austin et al., 2021), introducing concepts like uniform transitions, used in DiGress (Vignac et al., 2022) with node and edge labels, and absorbing states diffusion, adopted in GraphARM (Kong et al., 2023) for masking nodes.

## 3 REMOVING NODES AS A GRAPH NOISE PROCESS

In this work, we frame the process of removing nodes from a graph $\mathcal{G}$ as a diffusion process, which gradually corrupts $\mathcal{G}$ until there is no information left of the original graph. This can be interpreted

as an absorbing state diffusion process with node masks (Austin et al., 2021; Kong et al., 2023). Differently from (Kong et al., 2023), we do not limit the process to the choice of one node per step, but we study the diffusion process both from the node ordering and nodes number perspectives. In this sense, it can be said that such a process adds noise to $\mathcal{G}$ until it reaches the absorbing state of the empty graph $\varnothing$ after $T$ steps.

We define the node removal process as a noise process on graphs that randomly removes nodes at each step until no more are left. Given a graph data point $\mathcal{G}$, the removal process forms a removal sequence $\mathcal{G}_{0:T}^{\rightarrow}$ with $\mathcal{G}_0 = \mathcal{G}$ and $\mathcal{G}_T$ being the empty graph. We define the Markov removal transition $q(\mathcal{G}_t|\mathcal{G}_{t-1})$ as the probability of sampling a set of nodes $\mathcal{V}_t \subseteq \mathcal{V}_{t-1}$, and computing the induced subgraph $\mathcal{G}_t$ from $\mathcal{G}_{t-1}$ by $\mathcal{V}_t$. Following from Eq. 3, the forward process is defined as:

$$q(\mathcal{G}_{1:T}^{\rightarrow}|\mathcal{G}_0) = \prod_{t=1}^{T} q(\mathcal{G}_t|\mathcal{G}_{t-1}). \tag{5}$$

Now we show the key insight that, because $n_t = |\mathcal{V}_t|$ is a known property of $\mathcal{G}_t$, the removal transition can be broken down into two components:

$$q(\mathcal{G}_t|\mathcal{G}_{t-1}) = q(\mathcal{G}_t, n_t|\mathcal{G}_{t-1}) = q(\mathcal{G}_t|n_t, \mathcal{G}_{t-1})q(n_t|\mathcal{G}_{t-1}) \tag{6}$$

where $q(n_t|\mathcal{G}_{t-1})$ is the probability that $\mathcal{V}_t$ will have exactly $n_t$ nodes, and fixed this number, $q(\mathcal{G}_t|n_t, \mathcal{G}_{t-1})$ is the probability of choosing the nodes in $\mathcal{V}_t$ from $\mathcal{V}_{t-1}$. In simpler words, $q(n_t|\mathcal{G}_{t-1})$ tells *how many* nodes to keep alive, and once this fact is known, $q(\mathcal{G}_t|n_t, \mathcal{G}_{t-1})$ tells *which* nodes to keep alive. In some special cases of the removal process, we will show that the number of nodes $n_{t-1}$ is actually enough information to sample $n_t$, i.e., $q(n_t|\mathcal{G}_{t-1}) = q(n_t|n_{t-1})$.

## 3.1 PARAMETERIZING THE REVERSE OF THE REMOVAL PROCESS

Again, following the theory of diffusion models (Section 2.1.1), we introduce the insertion process, which learns to regenerate the graphs corrupted by the removal process. Define $p_{\theta,\phi}(\mathcal{G}_{t-1}|\mathcal{G}_t)$ as the Markov insertion transition, which samples $r_t = n_{t-1} - n_t$ new nodes $\mathcal{V}_{t,B}$ given $\mathcal{G}_t$ with nodes $\mathcal{V}_{t,A} = \mathcal{V}_t$, together with edges $\mathcal{E}_{t,AB}, \mathcal{E}_{t,BA}$ connecting the two graphs. Then, through a merge operation (4), graph $\mathcal{G}_{t-1}$ is composed. The reverse process is defined as:

$$p_{\theta,\phi}(\mathcal{G}_{0:T}^{\rightarrow}) = \prod_{t=1}^{T} p_{\theta,\phi}(\mathcal{G}_{t-1}|\mathcal{G}_t) \tag{7}$$

where we omitted the $p_{\theta,\phi}(\mathcal{G}_T)$ term, as all the probability mass is already placed on the empty graph $\varnothing$. Again, we can break the transition into two components:

$$p_{\theta,\phi}(\mathcal{G}_{t-1}|\mathcal{G}_t) = p_{\theta,\phi}(\mathcal{G}_{t-1}, r_t|\mathcal{G}_t) = p_\theta(\mathcal{G}_{t-1}|r_t, \mathcal{G}_t)p_\phi(r_t|\mathcal{G}_t) \tag{8}$$

where we call $p_\phi(r_t|\mathcal{G}_t)$ the *insertion model*, with parameters $\phi$, and $p_\theta(\mathcal{G}_{t-1}|r_t, \mathcal{G}_t)$ the *filler model*, with parameters $\theta$. The role of each is respectively to: (1) given the current subgraph $\mathcal{G}_t$ decide how many nodes $r_t$ to add, (2) known this number and $\mathcal{G}_t$, generate the content of the new nodes and respective edges, and how to connect them to $\mathcal{G}_t$. Expanding $p_{\theta,\phi}(\mathcal{G}_{t-1}, r_t|\mathcal{G}_t)$, we have:

$$p_{\theta,\phi}(\mathcal{G}_{t,A}, \mathcal{G}_{t,B}, \mathcal{E}_{t,AB}, \mathcal{E}_{t,BA}, r_t|\mathcal{G}_{t,A}) = p_\theta(\mathcal{G}_{t,B}, \mathcal{E}_{t,AB}, \mathcal{E}_{t,BA}|r_t, \mathcal{G}_{t,A})p_\phi(r_t|\mathcal{G}_{t,A})$$
$$= p_\theta(\mathcal{W}_t|r_t, \mathcal{G}_t)p_\phi(r_t|\mathcal{G}_t) \tag{9}$$

where we packed the tuple $\mathcal{W}_t = (\mathcal{G}_{t,B}, \mathcal{E}_{t,AB}, \mathcal{E}_{t,BA})$ for brevity. In Appendix C, we show that when $q(r_t|\mathcal{G}_t, \mathcal{G}_0)$ and $q(\mathcal{W}_t|r_t, \mathcal{G}_t, \mathcal{G}_0)$ can be expressed in closed form, then they can be estimated by minimizing the variational upper bound:

$$L_{\text{vub}} = \mathbb{E}_{\mathcal{G}_0 \sim q(\mathcal{G}_0)}\left[ \sum_{t=2}^{T} D_{\text{KL}}\big(q(r_t|\mathcal{G}_t, \mathcal{G}_0)\|p_\phi(r_t|\mathcal{G}_t)\big) - \mathbb{E}_{\mathcal{G}_1 \sim q(\mathcal{G}_1|\mathcal{G}_0)}\left[\log p_\phi(r_1|\mathcal{G}_1)\right] + \right.$$
$$\left. + \sum_{t=2}^{T} D_{\text{KL}}\big(q(\mathcal{W}_t|r_t, \mathcal{G}_t, \mathcal{G}_0)\|p_\theta(\mathcal{W}_t|r_t, \mathcal{G}_t)\big) - \mathbb{E}_{\mathcal{G}_1 \sim q(\mathcal{G}_1|\mathcal{G}_0)}\left[\log p_\theta(\mathcal{W}_1|r_1, \mathcal{G}_1)\right] \right]. \tag{10}$$

The KL divergence of the lower term can be replaced by the negative log-likelihood, at the price of increasing the upper bound. On the other hand, this allows to train $p_\theta(\mathcal{W}_t|r_t, \mathcal{G}_t)$ through any likelihood maximization method, such as VAE, Normalizing Flow, and Diffusion. Noticing the resemblance with Eq. 2, the filler model can be any likelihood-based one-shot model, although adapted to be conditioned on an external graph $\mathcal{G}_t$, and able to generate the interconnections (see Section 4.2.)

### 3.2 Choosing the removal process

This new framework is general enough to model $q(\mathcal{G}_t|\mathcal{G}_{t-1})$, and in turn the graph generative model $p_{\theta,\phi}(\mathcal{G}_{t-1}|\mathcal{G}_t)$, in several ways. We start with a naive *coin flip* approach to removing nodes. Then, we explore a more effective way of choosing the number of nodes to remove, and how to incorporate node ordering. All proofs are in Appendix C.

**Naive/binomial (Appendix B.1)** The simplest way to remove nodes from a graph $\mathcal{G}_{t-1}$ is to assign a Bernoulli random variable with probability $q_t$ to each node. All nodes with a positive outcome are removed. Then, given that at step $t-1$ the graph has $n_{t-1}$ nodes, the count of survived nodes $n_t$ is distributed as a Binomial $B(n_t; n_{t-1}, 1 - q_t)$. Iterating this process for $t$ steps from graph $\mathcal{G}_0$, we still get that $n_t|n_0$ is distributed as a Binomial, where the probability of being alive at step $t$ is the product of being alive at all steps. Finally, the posterior $q(\mathcal{G}_t|n_t, \mathcal{G}_{t-1})$, needed for computing the loss, is again distributed as a Binomial on the removed nodes $\Delta n_t = n_0 - n_t$.

**Categorical (Appendix B.2)** One drawback of the binomial removal is that, in principle, any block size can be sampled. This can be a problem when batching multiple examples (see Section 4.3), and leads to a big variance in the training examples. With the aim of controlling the size of blocks generated, and limiting the options available to the model, we develop a categorical removal process where the insertion model can choose from a predefined set of options. We base the formulation on the change-making problem (Wright, 1975), interpreting the number of nodes as the amount to be made using a set of coin denominations $D$, which will be the possible choices of the insertion model. Then, a removal transition is defined on $D$ as the frequencies in which each coin is used to make $n$ with the lowest amount of coins.

**Node ordering (Appendix B.3)** Until now, we assumed the nodes were removed in a uniform order in all the possible permutations. This doesn't need to be the case, as the whole removal process can be conditioned on a particular node ordering $\pi$. The transitions will then be of the form:

$$q(\mathcal{G}_t|\mathcal{G}_{t-1}, \pi) = q(\mathcal{G}_t|n_t, \mathcal{G}_{t-1}, \pi)q(n_t|\mathcal{G}_{t-1}, \pi) = q(n_t|\mathcal{G}_{t-1}, \pi). \tag{11}$$

**Halting process** Although a generative model could be complete with only the insertion and filler model, it still requires knowing when to stop. One possibility would be to stop after a fixed number of steps $T$, or better when some property of the removal process is met (e.g., the predicted number of remaining nodes $\Delta n_t$ is 0 for the binomial removals). To generalize to any case, we include an halting process (as defined in 6), which is set to 1 for the true data graph $\mathcal{G}$, and 0 for all its induced subgraphs. The halting signal is learned by a halting model $\lambda_\nu(\mathcal{G}_s, s)$ (binary classification task).

## 4 Uncovering the spectrum of sequentiality

Sequential and one-shot graph generative models (Section 2.1) are seen as two different families of graph generative models. Here we show that these are actually the two extremes of a spectrum, captured by our Insert-Fill-Halt (IFH) model (Figure 1). First of all, let's consider the reversed removal sequence (Definition 5) $\mathcal{G}_{0:T}^\leftarrow = (\mathcal{G}_s)_{s=0}^T$. Then the three modules are: (1) a *Node Insertion Module* $p_\phi(r_{s-1}|\mathcal{G}_{s-1})$ decides how many nodes are going to be inserted to $\mathcal{G}_{s-1}$; (2) a *Filler Module* $p_\theta(\mathcal{W}_{s-1}|r_{s-1}, \mathcal{G}_{s-1})$ fills the new $r_{s-1}$ nodes and edges $\mathcal{W}_{s-1}$, which are then merged with the existing graph $\mathcal{G}_{s-1}$ to get $\mathcal{G}_s$; (3) a *Halting Module* $\lambda_\nu(\mathcal{G}_s)$ decides, through some halting criteria, whether to stop the generative process at $s$ or to continue. The overall model distribution is:

$$p_{\theta,\phi,\nu}(\mathcal{G}) = \sum_{T=1}^\infty \sum_{\mathcal{G}_{0:T}^\leftarrow \in \mathcal{R}(\mathcal{G},T)} \underbrace{\lambda_\nu(G_T)}_{\text{halt at last step}} p_\theta(\mathcal{W}_{T-1}|r_{T-1}, \mathcal{G}_{T-1}) p_\phi(r_{T-1}|\mathcal{G}_{T-1}) \tag{12}$$

$$\prod_{s=1}^{T-1} \underbrace{(1 - \lambda_\nu(\mathcal{G}_s))}_{\text{do not halt}} \underbrace{p_\theta(\mathcal{W}_{s-1}|r_{s-1}, \mathcal{G}_{s-1})}_{\text{fill}} \underbrace{p_\phi(r_{s-1}|\mathcal{G}_{s-1})}_{\text{insert}}. \tag{13}$$

### 4.1 Specializing to one-shot and sequential models

**One-shot** One-shot models (Equation 2) are 1-step instances of our IFH model with the insertion module set to be a sampler of the total number of nodes, i.e., $p_\phi(r_0|\varnothing) = p_\phi(n_1) = p_\phi(n)$. The filler model is the actual one-shot model. The halting model always stops in 1 step.

**Sequential** Sequential models (Equation 1) are $n$-step instances of our IFH model, with the insertion

module always choosing 1 as the nodes to insert. The filler model samples a new node and links it with graph $\mathcal{G}_{s-1}$ to compose $\mathcal{G}_s$. The halting model is dependent on the architecture: in You et al. (2018), an End-Of-Sequence (EOS) token is sampled to end generation; in Liao et al. (2019) and Shi et al. (2020) it is not clear, but we assume they fix the number of nodes at the start; in Luo et al. (2021) generation stops when a limit on $n$ is reached, or if the model does not link the newly generated node to the previous subgraph; Han et al. (2023) trains a neural network to predict the halting signal from the adjacency matrix.

## 4.2 ADAPTING ONE-SHOT MODELS TO SEQUENTIAL

In Section 4.1, we showed how one-shot models are 1-step IFH models, and our parametrization in Section 3.1 allows the use of any one-shot model inside a multi-step instance. However, the model needs also to generate the edges linking new nodes with the previous subgraph, and to condition the former on the latter. Let $\mathcal{G}_{s-1}$ be the already generated subgraph, and $\mathcal{W}_{s-1}$ the new subgraph and inter-connections. We propose the following adaptation to the $T$-step setup for undirected graphs: (1) encode the $n_{s-1}$ nodes of graph $\mathcal{G}_{s-1}$ through a Graph Neural Network such as GraphConv (Morris et al., 2019) or RGCN (Schlichtkrull et al., 2018) for labeled data; (2) generate the new $r_{s-1}$ nodes using the encoded nodes, and a rectangular adjacency matrix with size $r_{s-1} \times n_s$, where $n_s = r_{s-1} + n_{s-1}$; (3) merge $\mathcal{G}_{s-1}$ and $\mathcal{W}_{s-1}$ into $\mathcal{G}_s$ by concatenating node and edge features.

## 4.3 COMPLEXITY CONSIDERATIONS

One-shot models generating adjacency matrices have a quadratic dependence on the number of nodes for both time and memory. However, through parallelizable computing architectures such as GPUs, it is possible to compute all components at the same time. It is not the case for autoregressive models where, due to their iterative nature, they cannot benefit from parallelization (Liao et al., 2019). Still, these do not need to generate the whole adjacency matrix in one go, and can better handle the already-generated graph representation, e.g., converting to a sparse representation. Another factor that affects memory and time is *batching*, that is, generating or training on many graphs at the same time, stacking their features in tensors. For dense representations, like adjacency matrices, the size of the resulting batched tensor always follows the biggest of the batch, and the rest have masked components. For a one-shot model then, batched adjacency matrices always have shape $n_{\max} \times n_{\max}$, where $n_{\max}$ is the maximum number of nodes in the batch. On the other hand, sequential models, especially those that generate one node at each step, do not suffer from this problem, as the graphs can be efficiently represented using one single adjacency list (this is also implemented in Torch Geometric Fey & Lenssen (2019)). Still, when parallelizing training of autoregressive models on all steps, the price is paid by replicating the same example many times, just with masked nodes. We show empirically in Section 5 that these considerations are confirmed in reality.

## 5 EXPERIMENTS

We experimentally evaluate how changing the formulation of the removal process changes sample quality and sampling time/memory consumption. Liao et al. (2019) already performed a sample quality/time trade-off analysis on a grid graphs dataset, changing the fixed block size, stride, and node ordering. We extend this analysis to a multitude of molecular and generic graph datasets, evaluating different degrees of sequentiality, i.e., size of sampled block sizes, on sample quality/time/memory usage. To showcase our framework, we adapt DiGress (Vignac et al., 2022) following the procedure in 4.2. Here we focus on domain-agnostic learning. Our method can be applied as-is to any graph dataset without needing additional information, apart from one-shot variants that need the node frequencies (see Section 2.1). Thus, we use the base version of DiGress, without optimal prior and domain-specific features. We only include nodes in-degrees and the number of nodes as additional features. Additional details on experiments are provided in Appendix D.

**Datasets** In the main text, we report results on two of the most popular molecular datasets: QM9 (Ramakrishnan et al., 2014), and ZINC250k (Irwin et al., 2012) with 133K and 250K molecules, respectively. Results on generic graph datasets are reported in Appendix A.1. As usual, we kekulize the molecules, i.e., remove the hydrogen atoms and replace aromatic bonds with single and double bonds, using the chemistry library RDKit (Landrum et al.). To measure sample quality we follow

Table 1: QM9 selection study for binomial vs. categorical removal, uniform vs. BFS ordering.

| Method | Valid (%)↑ | Unique (%)↑ | Novel (%)↑ | NSPDK↓ | FCD↓ | Time (m) | Memory (GB) |
|--------|-----------|-------------|------------|--------|------|----------|-------------|
| bin unif | 93.77 | 97.56 | 93.38 | 6.7e-4 | 1.132 | 60.96 | 2.11 |
| bin BFS | 92.73 | 96.85 | 91.11 | 8.2e-4 | 0.980 | 63.13 | 2.15 |
| cat unif | 93.71 | 97.86 | 92.30 | 7.7e.4 | 1.082 | 25.22 | 0.83 |
| cat BFS | 95.59 | 96.77 | 88.02 | 9.2e-4 | 0.893 | 25.54 | 0.83 |

the approach in Huang et al. (2023) and compute the Fréchet ChemNet Distance (FCD) and Neighborhood Subgraph Pairwise Distance Kernel (NSPDK) metrics. We also compute the ratio of Valid, Unique, and Novel molecules. Following Vignac et al. (2022), we report validity of molecules allowing partial charges. For both datasets, we generate 10K molecules, and evaluate FCD and NSPDK on the respective test sets: 10K molecules for QM9 and the canonical 25K test set for ZINC250k. For validation, we use another 10K molecules from QM9 and 10% of the ZINC250k training set.

## 5.1 EXPERIMENTAL RESULTS

**Selection study** We conducted a preliminary selection study on QM9 (shown in Table 1) to evaluate the best-performing formulation for the removal process from those we proposed. At the same time, this can be seen as an ablation study showing the contribution of more naive orderings and removal processes compared to better-engineered ones. For binomial removals, we used the adaptive linear scheduling B.1.1, and for categorical removals we used $D = \{1, 4\}$ as block sizes. As predicted in Section 4.3, the model trained with binomial removals has a huge memory footprint and worse sampling time compared to categorical removal. Coupled with the slightly worse sample quality, we see that the categorical removal process is superior. Regarding the ordering, BFS is able to improve quality compared to uniformly random order, as also observed in Liao et al. (2019).

**Baselines** For each dataset we define 4 degrees of sequentiality of our model (Appendix D). On molecular datasets we compare with most of the baselines from Huang et al. (2023), which are many state-of-the-art models. In particular: the autoregressive models GraphAF (Shi et al., 2020), GraphDF (Luo et al., 2021), GraphARM (Kong et al., 2023); the one-shot models MoFlow (Zang & Wang, 2020), EDP-GNN (Niu et al., 2020), GraphEBM (Liu et al., 2021), GDSS (Jo et al., 2022), CDGS (Huang et al., 2023). Results can be found in Table 2.

**Performance of the spectrum** In Table 2 we can see that the fully sequential model achieves competitive results with CDGS, and surpasses all autoregressive baselines on both QM9 and ZINC250k, where in the latter is even able to reach state-of-the-art validity and FCD. Moving from sequential down shows a drop in overall performance, although the one-shot variant restores a good validity in QM9. We also tried to increase (almost double) the number of parameters in the one-shot+ variant for ZINC250k, greatly surpassing the number of parameters of the autoregressive variants. Even with this advantage, the one-shot model was still not able to keep up.

**Time and memory consumption** Increasing the level of sequentiality towards 1-node sequential always seems to reduce the memory footprint during generation, as smaller and smaller adjacency matrices are generated, but time goes up, as predicted in Section 4.3. At the same time, we see that for small graphs datasets such as QM9, memory usage during training is higher in sequential models, differently from bigger sizes graph datasets like ZINC, where the cost of storing big adjacency matrices starts to outweigh that of split sparse graphs. Finally, even though the total training time increases with a higher sequentiality, models converge faster in wall-clock time.

**Generic graphs datasets** We also discuss the results of our model on generic graph datasets, which can be found in Table 3 in Appendix A due to the page limit. Here, we observe the same improvement increasing the sequentiality of the model. Concerning memory consumption we highlight the case with the Enzymes and Ego datasets, which contain very large graphs. On these datasets the sequential model uses respectively 1/50 and 1/88 of the memory footprint of the one-shot model for generation, although with an increased computational time. Surprisingly, we found there are minima in time and memory along the spectrum: for Ego, the block-sequential approach with big blocks is 5.5 and 2.4 times faster than the 1-node sequential and one-shot models, respectively.

Table 2: Results on the molecule generation task on QM9 (a-c) and ZINC250k (d-f). Tables on the left report performance results, while the tables on the right show the time/memory cost for different levels of sequentiality. On the comparison tables, the best results are in bold, and the second best are underlined. The training time is "time to reach best validation performance (all epochs time)".

(a) Performance results on the QM9 dataset

|  | Method | Valid (%)↑ | NSPDK↓ | FCD↓ | Unique (%)↑ | Novel (%)↑ |
|---|---|---|---|---|---|---|
|  | Train | - | 1.36e-4 | 0.057 | - | - |
| Autoreg. | GraphAF | 74.43 | 0.021 | 5.625 | 88.64 | 86.59 |
|  | GraphDF | 93.88 | 0.064 | 10.928 | 98.58 | 98.54 |
|  | GraphARM | 90.25 | 0.002 | 1.220 | 95.62 | 70.39 |
| One-shot | MoFlow | 91.36 | 0.017 | 4.467 | 98.65 | 94.72 |
|  | EDP-GNN | 47.52 | 0.005 | 2.680 | 99.25 | 86.58 |
|  | GraphEBM | 8.22 | 0.030 | 6.143 | 97.90 | 97.01 |
|  | DiGress | 99.00 | 5e-4 | 0.360 | 96.66 | 33.40 |
|  | GDSS | 95.72 | 0.003 | 2.900 | 98.46 | 86.27 |
|  | CDGS | 99.54 | **3.7e-4** | **0.269** | 97.20 | 72.52 |
| Ours | seq-1 | **99.96** | 0.001 | 0.876 | 96.24 | 85.00 |
|  | {1, 2} | 96.52 | 9.0e-4 | 0.896 | 96.55 | 86.94 |
|  | {1, 4} | 95.59 | 9.2e-4 | 0.894 | 96.77 | 88.02 |
|  | one-shot | 99.36 | 0.001 | 0.897 | 96.22 | 88.86 |

(b) Training time/memory

| Method | Time (h) | Memory (GB) |
|---|---|---|
| seq-1 | 2.9 (64) | 6.52 |
| {1, 2} | 28 (59) | 5.40 |
| {1, 4} | 40 (56) | 6.05 |
| one-shot | 4.6 (33) | 3.73 |

(c) Generation time/memory

| Method | Time (m) | Memory (GB) |
|---|---|---|
| seq-1 | 23.30 | 0.38 |
| {1, 2} | 20.55 | 0.48 |
| {1, 4} | 25.54 | 0.83 |
| one-shot | 16.92 | 1.22 |

(d) Performance results on the ZINC250K dataset

|  | Method | Valid (%)↑ | NSPDK↓ | FCD↓ | Unique (%)↑ | Novel (%)↑ |
|---|---|---|---|---|---|---|
|  | Train | - | 5.91e-5 | 0.985 | - | - |
| Autoreg. | GraphAF | 68.47 | 0.044 | 16.023 | 98.64 | 99.99 |
|  | GraphDF | 90.61 | 0.177 | 33.546 | 99.63 | 100.00 |
|  | GraphARM | 88.23 | 0.055 | 16.260 | 99.46 | 100.00 |
| One-shot | MoFlow | 63.11 | 0.046 | 20.931 | 99.99 | 100.00 |
|  | EDP-GNN | 82.97 | 0.049 | 16.737 | 99.79 | 100.00 |
|  | GraphEBM | 5.29 | 0.212 | 35.471 | 98.79 | 100.00 |
|  | DiGress | 91.02 | 0.082 | 23.06 | 81.23 | 100.00 |
|  | GDSS | 97.01 | 0.019 | 14.656 | 99.64 | 100.00 |
|  | CDGS | 98.13 | **7.03e-4** | 2.069 | 99.99 | 99.99 |
| Ours | seq-1 | **98.59** | 0.056 | **1.592** | 99.98 | 99.92 |
|  | {1, 3} | 85.56 | 0.057 | 2.173 | 99.99 | 99.93 |
|  | {1, 4, 8} | 80.67 | 0.051 | 5.202 | 100.00 | 99.97 |
|  | one-shot | 81.41 | 0.047 | 8.386 | 100.00 | 99.99 |
|  | one-shot+ | 83.82 | 0.052 | 7.028 | 99.96 | 99.99 |

(e) Training time/memory

| Method | Time (h) | Memory (GB) |
|---|---|---|
| seq-1 | 27 (127) | 30.17 |
| {1, 3} | 39 (86) | 33.06 |
| {1, 4, 8} | 52 (87) | 30.73 |
| one-shot | 24 (66) | 39.12 |
| one-shot+ | 46 (80) | 39.86 |

(f) Generation time/memory

| Method | Time (m) | Memory (GB) |
|---|---|---|
| seq-1 | 51.09 | 0.59 |
| {1, 3} | 26.71 | 1.08 |
| {1, 4, 8} | 36.39 | 3.05 |
| one-shot | 44.43 | 18.03 |
| one-shot+ | 58.04 | 19.22 |

# 6 DISCUSSION

In Section 5 we showed that sequentiality is directly linked with improved performances for our chosen filler model. At the same time, one can trade off generation time for memory and performance, although for bigger graphs there seems to be a sweet spot inside the spectrum. This indicates that the optimal removal process is dataset and task dependent, and could be considered as a hyperparameter when investigating new graph generative models. Our conjecture is that for smaller graph datasets, one-shot models are the fastest and best-performing solution, while as size increases, sequential models should be the go-to, particularly where memory is highly constrained. There is still room for improvement on this work's current limitations: for instance, designing better schemes to learn the halting model can be beneficial, as bigger graphs mean sparser halting signals to train on.

# 7 CONCLUSION

In this work, we proposed the IFH framework, which unifies the one-shot and autoregressive paradigms. We showed a new way of designing autoregressive models with a few examples, adapting strong one-shot models to act as their core, and reaching competitive results in a domain-agnostic setup. We want our work to be a stepping stone into building high-performing autoregressive graph generative models, which are capable of learning every aspect of a graph dataset, from the connectivity to the size of graphs. Doing so would unlock the opportunity for conditioning properties to fully control how big and how connected a graph might be.

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

## A  GENERIC GRAPHS GENERATION

### A.1  DATASETS

On generic graphs we follow Huang et al. (2023) and Vignac et al. (2022), and evaluate on: Community-small, with 100 graphs (You et al., 2018); Ego-small and Ego with 200 and 757 graphs (Sen et al., 2008); Enzymes with 563 protein graphs (Schomburg et al., 2004). We split the train/validation/test sets with the 60/20/20 proportion. For Community-small we compute the ratios of the MMDs with Gaussian EMD kernel for the distributions of Degree (Deg.), Clustering coefficient (Clus.) and Orbit, whereas for the remainder of datasets we strictly follow Huang et al. (2023) and compute the MMD with radial basis on the distribution of Degree, Clustering coefficient, Laplacian Spectrum coefficient (Spec.) and random GIN embeddings (Thompson et al., 2022), which are a replacement of FCD for generic graphs. For Ego-small we generate 1024 graphs, and for Community-small, Ego and Enzymes we generate the same number of graphs as the test set, which puts us in the same setups of Huang et al. (2023) and Vignac et al. (2022).

On Community-small we compare with GraphRNN (You et al., 2018), GRAN (Liao et al., 2019) and standard DiGress (Vignac et al., 2022) (Table 3b). On the remainder of generic graph datasets we compare with the autoregressive models GraphAF (Shi et al., 2020), GraphDF (Luo et al., 2021), GraphARM (Kong et al., 2023); the one-shot models MoFlow (Zang & Wang, 2020), EDP-GNN (Niu et al., 2020), GraphEBM (Liu et al., 2021), GDSS (Jo et al., 2022), CDGS (Huang et al., 2023) (Table 3). Results are discussed in Section 5.1 of the main text.

### A.2  INVESTIGATED LEVELS OF SEQUENTIALITY

In Table 3a we show our chosen levels of sequentiality, starting from 1-node sequential, then small blocks, then big blocks (also with different sizes), and finally one-shot with $n$ sampled from the dataset empirical distribution on number of nodes. We chose bigger coin denominations for Ego in the seq-big variant, as it contains much larger graphs. Notice that using the categorical removal process (Section B.2), having biggest coin 2 will roughly reduce the number of steps by two times with respect to 1-node sequential, and so on.

### A.3  DETAILED DISCUSSION ON RESULTS

In this section, we expand our findings on generic graphs datasets, which are presented in Table 3. The strong one-shot baseline CDGS surpasses every instance of our model in all but one metric. However, our seq-1 model still manages to beat all autoregressive baselines, and reaching second place for many metrics. We argue the performance in these datasets can be improved with a better halting mechanism: particularly for seq-1, the halting signal for training is very sparse. Think of a graph with 500 nodes from Ego, it means that the halting model is trained to predict class 0 (continue) for 499 subgraphs, and class 1 (halt) for the original graph. The same reasoning can be applied to the insertion model, which is trained to use the biggest block size most of the time.

Memory usage in generation is always improved by increasing sequentiality, while for training it seems to be quite stable. The latter is due to the balancing between the quadratic cost of adjacency matrices, and splitting across steps with smaller block sizes (also discussed in section 4.3).

Regarding computational time, we observe that there exist dataset-specific minima. For example, in the Ego dataset with big graphs, seq-big takes the smallest time to run. This might be a sweet spot between how parallel a block generation can be, and the number of steps to generate. The same is observed in Enzymes, where the minimum seems to be between seq-small and seq-big.

## B  REMOVAL PROCESSES

In this section we provide further details on the removal processes introduced in Section 3.2. All proofs for the equations can be found in Section C.

Table 3: Generic graphs results. Note that datasets have different numbers of generated test graphs, so memory and time are not to be compared from one dataset to the other. Training time refers to the time to train for all epochs.

(a) Sequentiality levels: block sizes used

| Method | Ego-small | Enzymes | Ego | Comm-small |
|---|---|---|---|---|
| seq-1 | 1 | 1 | 1 | 1 |
| seq-small | $\{1, 2\}$ | $\{1, 3\}$ | $\{1, 3\}$ | $\{1, 2\}$ |
| seq-big | $\{1, 2, 8\}$ | $\{1, 2, 8\}$ | $\{1, 4, 16\}$ | $\{1, 2, 8\}$ |
| oneshot | $n$ | $n$ | $n$ | $n$ |

(b) Community-small results

| Method | Deg.↓ | Clus.↓ | Orbit↓ | Ratio↓ |
|---|---|---|---|---|
| GraphRNN | 4.0 | 1.7 | 4.0 | 3.2 |
| GRAN | 3.0 | 1.6 | 1.0 | 1.9 |
| GG-GAN | 4.0 | 3.1 | 8.0 | 5.5 |
| SPECTRE | **0.5** | 2.7 | 2.0 | 1.7 |
| DiGress | 1.0 | **0.9** | **1.0** | **1.0** |
| seq-1 | 4.1 | **0.9** | 1.8 | 2.3 |
| seq-small | 5.9 | 1.2 | 11.4 | 6.2 |
| seq-big | 8.5 | 1.9 | 18.1 | 9.5 |
| oneshot | 8.5 | 12.9 | 49.1 | 23.5 |

(c) Performance results on generic graphs datasets

| | Method | Ego-small $\|V\|_{\max} = 17,\ \|E\|_{\max} = 66$ $\|V\|_{\mathrm{avg}} \approx 6,\ \|E\|_{\mathrm{avg}} \approx 9$ | | | | Enzymes $\|V\|_{\max} = 125,\ \|E\|_{\max} = 149$ $\|V\|_{\mathrm{avg}} \approx 33,\ \|E\|_{\mathrm{avg}} \approx 63$ | | | | Ego $\|V\|_{\max} = 399,\ \|E\|_{\max} = 1071$ $\|V\|_{\mathrm{avg}} \approx 145,\ \|E\|_{\mathrm{avg}} \approx 335$ | | | |
|---|---|---|---|---|---|---|---|---|---|---|---|---|---|
| | | Deg.↓ | Clus.↓ | Spec.↓ | GIN↓ | Deg.↓ | Clus.↓ | Spec.↓ | GIN↓ | Deg.↓ | Clus.↓ | Spec.↓ | GIN↓ |
| | Train | 0.025 | 0.029 | 0.027 | 0.016 | 0.011 | 0.011 | 0.011 | 0.007 | 0.009 | 0.009 | 0.009 | 0.005 |
| A-R | GraphRNN | 0.155 | 0.229 | 0.167 | 0.472 | 0.397 | 0.302 | 0.260 | 1.495 | 0.140 | 0.755 | 0.316 | 1.283 |
| | GRAN | 0.096 | 0.072 | 0.095 | 0.106 | 0.215 | 0.147 | 0.034 | 0.069 | 0.594 | 0.425 | 1.025 | 0.244 |
| O-S | VGAE | 0.146 | 0.046 | 0.249 | 0.089 | 0.811 | 0.514 | 0.153 | 0.716 | 0.873 | 1.210 | 0.935 | 0.520 |
| | EDP-GNN | 0.026 | 0.032 | 0.037 | 0.031 | 0.120 | 0.644 | 0.070 | 0.119 | 0.553 | 0.605 | 0.374 | 0.295 |
| | GDSS | 0.041 | 0.036 | 0.041 | 0.041 | 0.118 | 0.071 | 0.053 | 0.028 | 0.314 | 0.776 | 0.097 | 0.156 |
| | CDGS | **0.025** | **0.031** | **0.033** | **0.025** | **0.048** | 0.070 | **0.033** | **0.024** | **0.036** | **0.075** | **0.026** | **0.026** |
| Ours | seq-1 | 0.048 | 0.051 | 0.045 | 0.043 | 0.051 | **0.052** | 0.041 | 0.124 | 0.071 | 0.234 | 0.045 | 0.191 |
| | seq-small | 0.046 | 0.050 | 0.046 | 0.055 | 0.057 | 0.148 | 0.069 | 0.21 | 0.262 | 0.697 | 0.225 | 0.350 |
| | seq-big | 0.041 | 0.047 | 0.044 | 0.045 | 0.108 | 0.226 | 0.117 | 0.374 | 0.168 | 0.541 | 0.165 | 0.260 |
| | oneshot | 0.050 | 0.056 | 0.046 | 0.035 | 0.195 | 0.298 | 0.054 | 0.136 | 0.454 | 0.687 | 0.708 | 0.326 |

(d) Training time/memory

| | Ego-small | | Enzymes | | Ego | | Community-small | |
|---|---|---|---|---|---|---|---|---|
| Method | Time (h) | Memory (GB) | Time (h) | Memory (GB) | Time (h) | Memory (GB) | Time (h) | Memory (GB) |
| seq-1 | 12.15 | 3.19 | 55.5 | 13.68 | 77.86 | 22.17 | 14.39 | 3.81 |
| seq-small | 9.27 | 3.22 | 28.81 | 13.15 | 77.80 | 21.97 | 10.75 | 3.21 |
| seq-big | 14.29 | 7.54 | 28.07 | 14.81 | 58.28 | 22.44 | 12.5 | 4.62 |
| oneshot | 8.17 | 4.23 | 28.14 | 14.90 | 33.62 | 22.35 | 7.13 | 4.47 |

(e) Generation time/memory

| | Ego-small | | Enzymes | | Ego | | Community-small | |
|---|---|---|---|---|---|---|---|---|
| Method | Time (m) | Memory (GB) | Time (m) | Memory (GB) | Time (m) | Memory (GB) | Time (m) | Memory (GB) |
| seq-1 | 4.98 | 0.17 | 31.39 | 0.15 | 458.89 | 0.13 | 7.30 | 0.25 |
| seq-small | 3.36 | 0.20 | 11.36 | 0.19 | 268.89 | 0.17 | 5.62 | 0.30 |
| seq-big | 9.57 | 0.89 | 11.39 | 0.37 | 83.19 | 0.36 | 13.68 | 1.16 |
| oneshot | 5.18 | 1.60 | 23.59 | 7.51 | 202.73 | 11.40 | 7.42 | 2.24 |

## B.1 NAIVE (BINOMIAL)

The presented naive method is equivalent to tossing a coin for each node, and removing it for some outcome. A Bernoulli random variable with probability $q_t$ is assigned to each node. All nodes with a positive outcome are removed. The two components in Eq. 6 are found to be:

$$q(n_t|\mathcal{G}_{t-1}) = q(n_t|n_{t-1}) = \binom{n_{t-1}}{n_t} q_t^{n_{t-1}-n_t}(1-q_t)^{n_t} \qquad q(\mathcal{G}_t|n_t, \mathcal{G}_{t-1}) = \frac{1}{\binom{n_{t-1}}{n_t}} \qquad (14)$$

that is, the conditional $n_t|n_{t-1}$ is a Binomial random variable $B(n_t; n_{t-1}, 1-q_t)$, and $\binom{n_{t-1}}{n_t}$ are all the ways of choosing $n_t$ nodes from a total of $n_{t-1}$. Furthermore, we can obtain the $t$-step marginal

and posterior distributions as:

$$q(n_t|\mathcal{G}_0) = B(n_t; n_0, \pi_t) \qquad\qquad q(\mathcal{G}_t|n_t, \mathcal{G}_0) = \frac{1}{\binom{n_0}{n_t}} \qquad \text{with } \pi_t = \prod_{k=1}^{t}(1 - q_k) \quad (15)$$

$$q(r_t|\mathcal{G}_t, \mathcal{G}_0) = B(r_t; \Delta n_t, 1 - \bar{q}_t) \quad q(\mathcal{G}_{t-1}|r_t, \mathcal{G}_t, \mathcal{G}_0) = \frac{1}{\binom{\Delta n_t}{r_t}} \quad \text{with } \bar{q}_t = 1 - \frac{1 - \pi_{t-1}}{1 - \pi_t} \quad (16)$$

where $\Delta n_t = n_0 - n_t$ is the number of removed nodes from step $0$ to step $t$, and as such, can be reinserted to get back $\mathcal{G}_{t-1}$. The proofs for the equations are found in Section C.2. Loss 10 can't be used as it is because there are no reverse distributions for which the KL divergence can be computed without knowing $\Delta n_t$. This is because the support of a Binomial random variable is described by $\Delta n_t$, an information which is not available to the model. For this reason we follow the approach in Austin et al. (2021) and train the insertion model to predict $\Delta n_t$ from $\mathcal{G}_t$ through an MSE loss, and apply Eq. 16 for sampling.

The hyperparameters $q_t, \pi_t, \bar{q}_t$ can be defined as a schedule on $t$ (Ho et al., 2020). In particular we formulate the schedule in terms of $\pi_t$, which is the average ratio of alive nodes $n_t$ to total nodes $n_0$. We define a linear decay on $\pi_t$:

$$\pi_t = 1 - \frac{t}{T} \tag{17}$$

where $T$ is the number of removal steps as an hyperparameter. At time $t = 0$, all nodes are alive ($\pi_0 = 1$); at time $t = T/2$, half the nodes are alive on average ($\pi_{T/2} = 1/2$); at time $t = T$, all nodes have deterministically been removed ($\pi_T = 0$). $q_t$ and $\bar{q}_t$ are derived from Equation 17:

$$q_t = 1 - \frac{\pi_t}{\pi_{t-1}} = \frac{1}{T - t + 1} \tag{18}$$

$$\bar{q}_t = 1 - \frac{1 - \pi_{t-1}}{1 - \pi_t} = \frac{1}{t} \tag{19}$$

### B.1.1 ADAPTIVE SCHEDULING

With the linear decay schedule, the sizes of blocks depend on the true number of nodes $n_0$, as on average $n_0/T$ nodes are generated. To drop this dependency we make $T$ depend on the number of nodes $n_0$. A way to do so in linear scheduling is by setting:

$$T = \frac{n_0}{v}, \qquad \pi_t = 1 - v\frac{t}{n_0} \tag{20}$$

where $v$ is the *velocity* hyperparameter. The larger it is, the faster the decay. With this definition, $v$ is also the average number of nodes removed per step, e.g., if a graph has 12 nodes, and $v = 3$, then the graph will become empty in $T = 4$ steps, removing on average 3 nodes at a time. The name velocity comes from the physical interpretation of equation 20 as a law of motion.

### B.2 CATEGORICAL

The categorical removal process is based on the change-making problem (Wright, 1975): let $D \subset \mathbb{N}^d$ denote a set of $d$ coin denominations and, given a total change $C$, we want to find the smallest number of coins needed for making up that amount. This problem can be solved in pseudo-polynomial time using dynamic programming, and knowing the number of coins needed to make up the number of nodes $n_0$ of a graph $\mathcal{G}_0$ allows to build the shortest possible trajectory $\mathcal{G}_{0:T}^{\rightarrow}$ using the block size options in $D$. In particular, the number of steps $T$ will always be the number of coins that make the amount $n_0$. To select the number of removed nodes it is enough to pick any permutation of the coins that make $n_0$. This process retains the Markov property because the optimal sequence of coins for $n_t$ is a part of the optimal sequence for $n_0$, if $n_t$ is obtained by any optimal sequence. Categorical transitions describe a distribution on the choices of $D$:

$$q(r_t|n_{t-1}) = \frac{h(n_{t-1})[r_t]}{T - t + 1} \tag{21}$$

where $h(n_{t-1})$ is the histogram on the number of coins in $D$ that make up the amount $n_{t-1}$, $h(n_{t-1})[r_t]$ is the entry corresponding to denomination $r_t$, and $T - t + 1$ is the normalization constant, and also the number of coins making up $n_{t-1}$. The $t$-step marginal and posterior distribution

can be obtained as:

$$q(n_t|n_0) = \frac{\prod_{d \in D} \binom{h(n_0)[d]}{h(\Delta n_t)[d]}}{\binom{T}{t}} \tag{22}$$

$$q(r_t|n_0, n_t) = \frac{h(\Delta n_t)[r_t]}{t} \tag{23}$$

where $n_t|n_0$ is a multivariate hypergeometric random variable, and $r_t|n_0, n_t$ has the same distribution form of $r_t|n_{t-1}$. The interpretation of the multivariate hypergeometric is that the coins are now colored balls, and an urn contains exactly each of these balls with histogram $h(n_0)$. We need to shave the amount $\Delta n_t$, so we have to pick exactly the number of balls of each color contained in $h(\Delta n_t)$. We pick $t$ balls from a total of $T$ in the urn.

### B.3 NODE ORDERING

Until now we assumed the nodes were removed in a uniformly random order, enforced by the $q(\mathcal{G}_t|n_t, \mathcal{G}_{t-1})$, selecting which nodes to keep alive. One example is given by the naive case in Appendix B.1, where nodes are selected uniformly. This doesn't need to be the case, as $q(\mathcal{G}_t|n_t, \mathcal{G}_{t-1})$ can actually be any other distribution. Furthermore, to enforce the Markov property once more, we can condition the removal sequence $\mathcal{G}_{0:T}^{\rightarrow}$ on a particular node ordering $\pi$ before starting the removal. The transitions will then be of the form:

$$q(\mathcal{G}_t|\mathcal{G}_{t-1}, \pi) = q(\mathcal{G}_t|n_t, \mathcal{G}_{t-1}, \pi)q(n_t|\mathcal{G}_{t-1}, \pi) = q(n_t|\mathcal{G}_{t-1}, \pi) \tag{24}$$

The ordering $\pi$ can be taken into account in loss 10 in the outer expectation. In that case, we have to sample both an example $\mathcal{G}_0$, and a node ordering $\pi$.

## C PROOFS

### C.1 PROOF OF THE VARIATIONAL LOWER BOUND 10

*Proof.* Recall the notation in 2. To simplify the notation we consider $\mathcal{F}(\mathcal{G})$ as the set of any forward removal sequence of $\mathcal{G}$. Start from the prior distribution of the model:

$$
\begin{aligned}
p_{\theta,\phi}(\mathcal{G}_0) &= \sum_{\mathcal{G}_{1:T}^{\rightarrow} \in \mathcal{F}(\mathcal{G}_0)} p_{\theta,\phi}(\mathcal{G}_{0:T}^{\rightarrow}) && \text{tot. prob. over trajectories } \mathcal{F}(\mathcal{G}_0) \\
&= \sum_{\mathcal{G}_{1:T}^{\rightarrow} \in \mathcal{F}(\mathcal{G}_0)} p_{\theta,\phi}(\mathcal{G}_{0:T}^{\rightarrow}) \frac{q(\mathcal{G}_{1:T}^{\rightarrow}|\mathcal{G}_0)}{q(\mathcal{G}_{1:T}^{\rightarrow}|\mathcal{G}_0)} && \text{importance sampling} \\
&= \sum_{\mathcal{G}_{1:T}^{\rightarrow} \in \mathcal{F}(\mathcal{G}_0)} q(\mathcal{G}_{1:T}^{\rightarrow}|\mathcal{G}_0) p_{\theta}(\mathcal{G}_T) \frac{p_{\theta,\phi}(\mathcal{G}_{0:T-1}|\mathcal{G}_T)}{q(\mathcal{G}_{1:T}^{\rightarrow}|\mathcal{G}_0)} && \\
&= \sum_{\mathcal{G}_{1:T}^{\rightarrow} \in \mathcal{F}(\mathcal{G}_0)} q(\mathcal{G}_{1:T}^{\rightarrow}|\mathcal{G}_0) p_{\theta}(\mathcal{G}_T) \prod_{t=1}^{T} \frac{p_{\theta,\phi}(\mathcal{G}_{t-1}|\mathcal{G}_t)}{q(\mathcal{G}_t|\mathcal{G}_{t-1})} && \text{Markov property} \\
&= \sum_{\mathcal{G}_{1:T}^{\rightarrow} \in \mathcal{F}(\mathcal{G}_0)} q(\mathcal{G}_{1:T}^{\rightarrow}|\mathcal{G}_0) \frac{p_{\theta}(\mathcal{G}_T)}{q(\mathcal{G}_T|\mathcal{G}_0)} && \\
&\quad p_{\theta,\phi}(\mathcal{G}_0|\mathcal{G}_1) \prod_{t=2}^{T} \frac{p_{\theta,\phi}(\mathcal{G}_{t-1}|\mathcal{G}_t)}{q(\mathcal{G}_{t-1}|\mathcal{G}_t, \mathcal{G}_0)} && \text{rewriting as posteriors}
\end{aligned}
$$

$$= \sum_{\mathcal{G}_{1:T}^{\rightarrow} \in \mathcal{F}(\mathcal{G}_0)} q(\mathcal{G}_{1:T}^{\rightarrow}|\mathcal{G}_0) \frac{p_\theta(\mathcal{G}_T)}{q(\mathcal{G}_T|\mathcal{G}_0)} p_\phi(n_0|\mathcal{G}_1) p_\theta(\mathcal{G}_0|n_0, \mathcal{G}_1)$$

$$\prod_{t=2}^{T} \frac{p_\phi(n_{t-1}|\mathcal{G}_t)}{q(n_{t-1}|\mathcal{G}_t, \mathcal{G}_0)} \frac{p_\theta(\mathcal{G}_{t-1}|n_{t-1}, \mathcal{G}_t)}{q(\mathcal{G}_{t-1}|n_{t-1}, \mathcal{G}_t, \mathcal{G}_0)} \qquad \text{expanding as in 6}$$

$$= \sum_{\mathcal{G}_{1:T}^{\rightarrow} \in \mathcal{F}(\mathcal{G}_0)} q(\mathcal{G}_{1:T}^{\rightarrow}|\mathcal{G}_0) \frac{p_\theta(\mathcal{G}_T)}{q(\mathcal{G}_T|\mathcal{G}_0)} p_\phi(r_1|\mathcal{G}_1) p_\theta(\mathcal{W}_1|r_1, G_1)$$

$$\prod_{t=2}^{T} \frac{p_\phi(r_t|\mathcal{G}_t)}{q(r_t|\mathcal{G}_t, \mathcal{G}_0)} \frac{p_\theta(\mathcal{W}_t|r_t, \mathcal{G}_t)}{q(\mathcal{W}_t|r_t, \mathcal{G}_t, \mathcal{G}_0)}$$

The Variational Upper Bound is found from the negative log likelihood through the Jensen Inequality:

$$\mathbb{E}_{q(\mathcal{G}_0)}[-\log p_{\theta,\phi}(\mathcal{G}_0)] \leq \mathbb{E}_{q(\mathcal{G}_0)} \left[ \sum_{t=2}^{T} D_{\mathrm{KL}}\big(q(r_t|\mathcal{G}_t, \mathcal{G}_0)\|p_\phi(r_t|\mathcal{G}_t)\big) - \mathbb{E}_{q(\mathcal{G}_1|\mathcal{G}_0)}\left[\log p_\phi(r_1|\mathcal{G}_1)\right] + \right.$$

$$\left. + \sum_{t=2}^{T} D_{\mathrm{KL}}\big(q(\mathcal{W}_t|r_t, \mathcal{G}_t, \mathcal{G}_0)\|p_\theta(\mathcal{W}_t|r_t, \mathcal{G}_t)\big) - \mathbb{E}_{q(\mathcal{G}_1|\mathcal{G}_0)}\left[\log p_\theta(\mathcal{W}_1|r_1, \mathcal{G}_1)\right] \right]$$

$\square$

## C.2 Binomial removal

### C.2.1 Proof of equation 15

*Proof.* Let's prove by induction. Let's consider the simple case for $n_1$:

$$q(n_1|n_0) = B(n_1; n_0, \pi_1)$$

with $\pi_1 = 1 - q_1$. This is true due to the definition of a transition 14.

Now, assume the property is true for $t-1$, that is, $n_{t-1}|n_0$ is a Binomial $B(n_{t-1}; n_0, \pi_{t-1})$. We know that $n_t|n_{t-1}$ is also a Binomial, which is the same as $n_t|n_{t-1}, n_0$ due to the Markov property. Let's recall what their distribution and parameters are:

$$n_t|n_{t-1}, n_0 \sim B(n_{t-1}|n_t, 1 - q_t)$$

$$n_{t-1}|n_0 \sim B(n_0, \pi_{t-1}) \qquad \pi_{t-1} = \prod_{k=1}^{t-1}(1 - q_k)$$

It can be proven that a Binomial conditioned on a Binomial is still a Binomial with probability the product of the two probabilities, and number of experiments the same as the conditioning binomial. From this fact $n_t|n_0$ is a Binomial:

$$n_t|n_0 \sim B(n_0, \pi_t) \qquad \pi_t = (1 - q_t)\pi_{t-1} = \prod_{k=1}^{t}(1 - q_k)$$

$\square$

### C.2.2 PROOF OF EQUATION 16

*Proof.* Let's compute the posterior:

$$q(n_{t-1}|n_t, n_0) = q(n_t|n_{t-1}) \frac{q(n_{t-1}|n_0)}{q(n_t|n_0)}$$

$$= \frac{n_{t-1}!}{n_t!(n_{t-1}-n_t)!}(1-q_t)^{n_t} q_t^{n_{t-1}-n_t} \frac{\frac{n_0!}{n_{t-1}!(n_0-n_{t-1})!}\pi_{t-1}^{n_{t-1}}(1-\pi_{t-1})^{n_0-n_{t-1}}}{\frac{n_0!}{n_t!(n_0-n_t)!}\pi_t^{n_t}(1-\pi_t)^{n_0-n_t}}$$

$$= \frac{(n_0-n_t)!}{(n_{t-1}-n_t)!(n_0-n_{t-1})!}\pi_{t-1}^{n_{t-1}-n_t} q_t^{n_{t-1}-n_t} \frac{(1-\pi_{t-1})^{n_0-n_{t-1}}}{(1-\pi_t)^{n_0-n_t}}$$

$$= \frac{(n_0-n_t)!}{(n_{t-1}-n_t)!(n_0-n_t-(n_{t-1}-n_t))!}\pi_{t-1}^{n_{t-1}-n_t}(1-\pi_{t-1})^{n_0-n_{t-1}}\frac{q_t^{n_{t-1}-n_t}}{(1-\pi_t)^{n_0-n_t}}$$

$$= \binom{n_0-n_t}{n_{t-1}-n_t}\left(q_t\frac{\pi_{t-1}}{1-\pi_t}\right)^{n_{t-1}-n_t}\left(q_t\frac{1-\pi_{t-1}}{1-\pi_t}\right)^{n_0-n_{t-1}}$$

$$(\text{another way}) = \binom{n_0-n_t}{n_0-n_{t-1}}\left(\frac{1-\pi_{t-1}}{1-\pi_t}\right)^{n_0-n_{t-1}}\left(1-\frac{1-\pi_{t-1}}{1-\pi_t}\right)^{n_{t-1}-n_t}$$

Finally, by substituing the number of failures at step $t$: $r_t = n_{t-1} + n_t$, or equivalently the number of nodes that should be inserted:

$$q(r_t|n_t, n_0) = \binom{n_0-n_t}{r_t}\left(q_t\frac{\pi_{t-1}}{1-\pi_t}\right)^{r_t}\left(\frac{1-\pi_{t-1}}{1-\pi_t}\right)^{n_0-n_t-r_t}$$

$\square$

## D IMPLEMENTATION DETAILS

We implemented our framework using PyTorch (Paszke et al., 2019), PyTorch Lightning (Falcon & The PyTorch Lightning team, 2019) and PyTorch Geometric (Fey & Lenssen, 2019). Our foundation was the DiGress implementation (Vignac et al., 2022) , which we heavily modified and partly reimplemented to generalize on many cases. We will make our code publicly available after the reviewing process, and for now it is available as supplementary material.

All our experiments and hyperparameters are available in our code as simple Hydra (Yadan, 2019) configuration files, and each was run for 3 different seeds. For each experiment we also report the time to sample the set of test generated graphs and the memory footprint. We ran ZINC250k experiments on an A100-40GB GPU, Ego experiments on an L4 GPU, and all other experiments on a T4 GPU.

We implemented the insertion model and halting model (when needed) as RGCN (Schlichtkrull et al., 2018) to tackle labelled datasets, and GraphConvs (Morris et al., 2019) for unlabelled datasets. We implemented the halting model in the same way.

### D.1 TRAINING AND GENERATION

Algorithm 1 shows how to train the IFH model: sequential operations in the inner while loop can be performed in parallel by first sampling the whole sequence, and then computing the gradients on the collected batch in one computation. Whenever one of the models has no parameters (e.g., insertion model for the 1-node sequential case), its gradient accumulation step can be skipped. Algorithm 2 expresses the generation procedure to return a sampled graph $\mathcal{G}_T$. It can be seen that it is a reflection of definition 13, and makes explicit the Insert, Fill, Halt operations. Notice the usage of the split operation during training and merge operation during generation.

Finally, graphs dubbed by $\mathcal{G}$ are kept in sparse representation, as explained in Section 4.3, while splits $\mathcal{W}$ are the only part in dense representation, when required by the filler model.

**Algorithm 1** Training

1:  **repeat**
2:      $\mathcal{G}_0 \sim q(\mathcal{G})$
3:      **while** $\mathcal{G}_{t-1} \neq \varnothing$ **do**
4:          $\mathcal{G}_t \sim q(\mathcal{G}_{t-1}|\mathcal{G}_t)$
5:          $r_t \leftarrow n_{t-1} - n_t$
6:          $\mathcal{W}_t \leftarrow \mathrm{split}(\mathcal{G}_{t-1}, \mathcal{G}_t)$
7:          $h_t \leftarrow \delta(t-1)$
8:          Accumulate gradients:

$$\nabla_\phi D_{\mathrm{KL}}\big(q(r_t|\mathcal{G}_t, \mathcal{G}_0) \| p_\phi(r_t|\mathcal{G}_t)\big)$$
$$\nabla_\theta D_{\mathrm{KL}}\big(q(\mathcal{W}_t|r_t, \mathcal{G}_t, \mathcal{G}_0) \| p_\theta(\mathcal{W}_t|r_t, \mathcal{G}_t)\big)$$
$$\nabla_\nu \mathcal{L}_{\mathrm{halt}}(h_t, \lambda_\nu(\mathcal{G}_{t-1}))$$

9:      **end while**
10:     Perform gradient descent step
11: **until** converged

**Algorithm 2** Generation

1:  $\mathcal{G}_0 \leftarrow \varnothing$
2:  **repeat**
3:      $r_s \sim p_\phi(r_s|\mathcal{G}_s)$
4:      $\mathcal{W}_s \sim p_\theta(\mathcal{W}_s|r_s, \mathcal{G}_s)$
5:      $\mathcal{G}_{s+1} \leftarrow \mathrm{merge}(\mathcal{G}_s, \mathcal{W}_s)$
6:      $h_s \sim \lambda_\nu(\mathcal{G}_{s+1})$
7:  **until** $h_s = 1$
8:  **return** $\mathcal{G}_T$

## D.2    ADAPTING DIGRESS

We briefly discuss how we adapted the DiGress model and architecture to act as a filler model. The nodes of the already generated graph are encoded through an RGCN or GraphConv, and are used as input in the graph transformer architecture (Dwivedi & Bresson, 2020), together with the vectors of noisy labels of the new nodes. Noisy edges are sampled both between new nodes, and also from new nodes to existing nodes. In a graph transformer layer, new nodes can attend both to themselves and old nodes, and mix with the information on edges, as is done in DiGress. Finally, the vectors of new nodes and edges are updated through the Feed Forward Networks of the transformer layer, while the encoded old nodes remain untouched. With this last consideration, one can encode the nodes of the already generated graph only once in a filler model call, and use them in all the DiGress denoising steps.

