# OpenReview forum: "Uncovering the Spectrum of Graph Generative Models: From One-Shot to Sequential"
_ICLR.cc/2024/Conference — Submitted to ICLR 2024_

### Official Review · Reviewer_gvwJ · 2023-10-31

**Soundness:** 3 good
**Presentation:** 3 good
**Contribution:** 3 good
**Rating:** 5
**Confidence:** 4

**Summary:**

This paper presents a diffusion-based graph generative model that unifies both one-shot and autoregressive generative models. The node removal process is conducted with a denoising diffusion model and the insertion reverses the process by predicting the number of nodes that have been removed. Setting the number of nodes from 1 to n enables the unification of one-shot and autoregressive generation.

**Strengths:**

1. This paper proposes the novel unification between one-shot and autoregressive graph generative models using diffusion models.
2. The introduction of flexible-sized block-wise generation for graph generation stands out as a noteworthy contribution.

**Weaknesses:**

1. Can the unification of one-shot and autoregressive graph generative models be a strong contribution? For instance, GRAN (Liao et al., 2019) can also be the unification between one-shot and autoregressive graph generative model by setting the block size as the number of nodes. What is the key difference of the work from GRAN except for the usage of diffusion models?
2. I wonder if it is proper to say the performance as the new state-of-the-art results as mentioned in the abstract. The FCD for QM9 and NSPDK for ZINC do not seem to be state-of-the-art results. Also, as the authors adapted DiGress, the performance comparison with DiGress can be meaningful.
3. Lack of detailed analysis on the sample quality-time trade-off. A more detailed analysis of the correlation between the sample quality and time (or memory consumption) is needed by comparing the one-shot and autoregressive versions of the IFH model.

**Questions:**

1. Which level of sequentiality did the authors use (I cannot find details in appx D.)? Does the degree of sequentiality imply the block size (or the number of steps)?
2. The generic graph generation results in appx B do not look good enough. Is there any particular reason that the model works okay for molecular graphs but not for non-attributed generic graphs?

---

> ### Author Response · Authors · 2023-11-22
> **Answer to reviewer gvwJ**
>
> We thank you for your review and for asking for further details. We are happy to resolve all doubts.
>
> **W1: Can the unification of one-shot and autoregressive graph generative models be a strong contribution? For instance, GRAN (Liao et al., 2019) can also be the unification between one-shot and autoregressive graph generative model by setting the block size as the number of nodes. What is the key difference of the work from GRAN except for the usage of diffusion models?**
>
> We lean towards a positive answer, as we not only unify the one-shot and autoregressive paradigms, but also show how one can be adapted to the other. We also provide a formal way of designing new graph generative models without having to strictly choose between autoregressive and one-shot: one could design a new one-shot model, and quickly try the same formulation with different levels of sequentiality, trading off computational time and memory usage, and looking at how it affects performances. In a way, we unlock the level of sequentiality as a hyperparameter that can be tuned in any model to further improve quality and resource factors.
>
> We think the key difference between our IFH framework and GRAN is flexibility: GRAN makes a lot of assumptions, such as fixing the number of generated nodes to the maximum number of nodes in a dataset (actually the biggest multiple of the fixed block size), and cannot capture the true distribution on the number of nodes of a dataset. For this reason, there is a bias towards large graphs, as the biggest connected component is picked, and smaller graphs are not sampled.
>
> **W2: I wonder if it is proper to say the performance as the new state-of-the-art results as mentioned in the abstract. The FCD for QM9 and NSPDK for ZINC do not seem to be state-of-the-art results. Also, as the authors adapted DiGress, the performance comparison with DiGress can be meaningful.**
>
> We agree that for some metrics, the best one-shot baseline CDGS outperforms our approach. However, our sequential instances improve by a significant margin the autoregressive state-of-the-art, as shown in Table 2, while being competitive with CDGS. We thank you for recommending adding DiGress as a baseline, and we did so in the revised manuscript.
>
> **W3: Lack of detailed analysis on the sample quality-time trade-off. A more detailed analysis of the correlation between the sample quality and time (or memory consumption) is needed by comparing the one-shot and autoregressive versions of the IFH model.**
>
> Although we explored the subject in Section 5.1, we might not have been too detailed. For this reason, we added a more thorough discussion on the sample quality/time/memory trade-off in the new Discussion section 6. To summarize, we noticed an increase in memory consumption and decreased computational time as we moved away from the 1-node sequential model, as we expected in Section 4.3.
>
> **Q1: Which level of sequentiality did the authors use (I cannot find details in appx D.)? Does the degree of sequentiality imply the block size (or the number of steps)?**
>
> Thank you for noticing. We have updated the appendix referring to generic graphs, and moved it as Appendix A to make it more accessible from the main text, moving theoretical parts after. In particular, we added the "Sequentiality levels" Table (3a). You are right, the degree (or level) of sequentiality implies the sizes of blocks used, which in turn affects the number of steps. Consider the following with 53 nodes: one-shot inserts all 53 nodes in 1 step; sequential with categorical removal with blocks {1, 2, 8} will generate (with any permutation) 6 blocks of 8 nodes, 2 blocks of 2, and 1 of 1, as the insertion model is trained to pick the largest block size if needed, minimizing the number of steps, which in this case is 9; 1-node sequential model will take 53 steps. Notice that, even adding a block size of 2 to 1-node sequential makes it roughly halve its generation steps.
>
> **Q2: The generic graph generation results in appx B do not look good enough. Is there any particular reason that the model works okay for molecular graphs but not for non-attributed generic graphs?**
>
> Although we do not reach state-of-the-art on all metrics against the very strong one-shot model CDGS, the results on generic graphs beat all autoregressive baselines, and are competitive with one-shot baselines. However, as you said, we also noticed a drop in performance for non-attributed generic graphs, and we think that would be a promising direction to explore. Particularly for 1-node sequential models applied to big-sized graphs, we noticed that the halting signal's sparsity hinders the halting model's training process. We address this limitation in the Discussion section (numbered 6 in the revised version).

---

### Official Review · Reviewer_yahf · 2023-11-01

**Soundness:** 3 good
**Presentation:** 3 good
**Contribution:** 3 good
**Rating:** 5
**Confidence:** 4

**Summary:**

The authors propose a generalization of deep graph generative models that results in a spectrum between one-shot models and sequential models. They take inspiration from diffusion model theory to train a model on the corruption of graphs (removal of blocks of the node) to learn how to insert multiple nodes and fill in edges. They adapt a diffusion-based one-shot model DiGress to their approach (1-node sequential) and show that it outperforms state-of-the-art on some datasets.

**Strengths:**

- The method unifies one-shot and sequential generation methods and opens up new opportunities for searching for new graph generation methods.
- Evaluation covers several datasets and metrics.

**Weaknesses:**

1. **Performance on other datasets** - The paper presents the evaluations on two datasets in the main content and three datasets in the appendix. While the proposed approach outperforms the state-of-the-art in the former two, multiple one-shot methods outperform the proposed approach. This undermines the impact of the new approach. Also, it is not clear why only the two datasets with good performance were shown in the main paper. How about other datasets that have been used in prior work, such as Grid, Protein, and 3D point-cloud?

1. While the method unlocks a spectrum between one-shot and sequential models, it does not present a way to choose one from the spectrum. How many nodes should be added per step? Is this a hyperparameter? The presented experiments show that seq is better. Does "seq" refer to 1 node per step? There are seq-small and seq-big in the Appendix, but none of the variations outperform CDGS except for one metric on one dataset.

1. **Presentation Issues** - While the writing is understandable, there are several presentation issues. For example:
    - Definition 6: "*An* halting process ... ." Also, I don't think the first sentence completely defines the halting process; the second sentence does. So, this should be rewritten.
    - Page 6: "On the other hand, ...  such as VAE, Normalizing Flow, Diffusion" is missing an *and*.

**Questions:**

Addressing the following would significantly improve my score
1. Among the five datasets presented, the proposed approach does not outperform other methods in majority of the metrics. Can the authors  justify the utility of their approach given these results?

1. How to select $r_s$ and what are the differences between seq, seq-small, and seq-big?

---

> ### Author Response · Authors · 2023-11-22
> **Answer to reviewer yahf**
>
> Thank you for your comments and for asking for additional motivation for our work. We find that clarifying this point is of utmost importance. Let us start by answering your questions.
>
> **Q1: Among the five datasets presented, the proposed approach does not outperform other methods in majority of the metrics. Can the authors justify the utility of their approach given these results?**
>
> While it is true that the strongest one-shot baseline CDGS outperforms our approach in generic graphs datasets, our experiments show that increasing the degree of sequentiality with the chosen one-shot model DiGress leads to state-of-the-art results with respect to autoregressive generation, while also being competitive, if not better in many metrics, in molecular generation.
>
> Metrics comparison versus other methods helps to answer whether our best instance of the IFH model can be useful for good quality generation. However, we argue that this is not the only criterion for choosing a good model, and through our sample quality/memory/time trade-off analysis we investigated other aspects. In particular, how the degree of sequentiality influences performance, computing time, and memory consumption. These latter aspects might be needed for some kinds of datasets, e.g. big sized graph datasets, where we showed an improvement of 50 times less memory usage.
>
> Conceptually, we think one of the main take-home messages of our work is that, when designing new graph generative models, be it one-shot or autoregressive, there is always a way to explore the whole spectrum with it, trading off memory, time, and quality, and perhaps push its native performances. To do so, one can use our mathematical framework, which can be a way of standardizing techniques as building blocks.
>
> **Q2: How to select $r_s$ and what are the differences between seq, seq-small, and seq-big?**
>
> Actually, we don't manually select $r_s$ but design the process that stochastically generates $r_s$. Your question can be answered by our sample quality/time/memory trade-off analysis, in which we showed a significant improvement when using smaller block sizes concerning generation quality and memory, but not in time (due to not being able to parallelize during generation as well as one-shot models). When increasing the block sizes, we get the inverse, even though for larger datasets, we see that having larger blocks, but not too large, improves quality and time, so there must be a sweet spot depending on the size of the graphs in the dataset. To summarize, our empirical advice is to use one-shot models for smaller graphs, and go towards 1-node sequential models for bigger graphs. When graphs get too big, having a small block size requires a solid halting process, because the halting signal becomes very sparse in a setup with a huge number of steps. For this reason, increasing the block size to reduce the number of steps can help. We reported these points in the new Discussion section (numbered 6).
>
> We thank you for inquiring about the meaning of the levels of sequentiality in generic graphs. We updated the appendix section to explain better what we are doing in those experiments.
>
>
> We also would like to answer the proposed weaknesses which we did not answer above.
>
> **It is not clear why only the two datasets with good performance were shown in the main paper.**
>
> We made this choice due to page limitations (as generic graphs datasets tables occupy quite a lot of space) and because the related work from which we took major inspiration for our experimental setup (CDGS) put much more emphasis on molecular datasets. To address this as best as possible, we moved the generic graphs section to Appendix A to make it more accessible from the main text.
>
> **How about other datasets that have been used in prior work, such as Grid, Protein, and 3D point-cloud?**
>
> Thank you for the suggestion. We see that these are the datasets used in GRAN, which are not so widely adopted in more recent works. Although we already provided a wide range of datasets for evaluating our approach, we will add experiments for the Grid dataset, which is the one where the ablation study for GRAN was conducted. We consider this as a way of comparing our approach to theirs, and see whether we get an improvement. We will get back with the results soon.
>
> **Update:** due to technical difficulties we were not able to finalize the additional experiments on the Grid dataset. However, we believe that our method has already been evaluated on a significantly wide array of popular datasets, among which molecular and generic graphs. To the best of our knowledge, the Grid dataset has only been used in GRAN.

---

### Official Review · Reviewer_H5VJ · 2023-11-01

**Soundness:** 3 good
**Presentation:** 3 good
**Contribution:** 2 fair
**Rating:** 6
**Confidence:** 3

**Summary:**

This paper unifies the one-shot and autoregressive graph generation methods into a diffusion framework and proves that these two methods are two extremes of the unified model. Specifically, in the forward phase, blocks, i.e., a set of nodes, are gradually removed as the noise increases. In the backward phase, blocks are gradually added as the denoising process proceeds. When the block size is set to 1, the diffusion model degenerates to an autoregressive approach. When the block size is equal to the graph size, the diffusion model becomes a one-shot method. Experiments on both molecular and generic graphs witness a trade-off between the quality and time of sampling.

**Strengths:**

1. This paper unifies the autoregressive and one-shot graph generation methods into a unified diffusion model, where the removal of nodes is used as the forward process and the generation of nodes is the denoising process. The idea is sound and interesting.

2. The proposed method trade-offs the quality and time of sampling. The proposed method outperforms state-of-the-art autoregressive methods when degenerating to 1-node sequential.

**Weaknesses:**

1. This paper combines the ideas of autoregressive graph diffusion [1] and block generation [2]. Although the combination is natural, I am not clear on the main difference between the proposed method and GRAN. It seems that the unity of autoregression and one-shot is due to the design of block generation, rather than the diffusion of node removal.

2. It's not clear to me what advantages 1-node IFH has over autoregressive methods. Does the benefit come from the prediction of the number of nodes?

3. The time and memory costs of baselines are not reported in Tables 2 and 3. It is therefore impossible to see the trade-off between sampling quality and time.

[1] Autoregressive Diffusion Model for Graph Generation. ICML 2023.

[2] Efficient Graph Generation with Graph Recurrent Attention Networks. NeurIPS 2019.

**Questions:**

See weaknesses.

---

> ### Author Response · Authors · 2023-11-22
> **Answer to reviewer H5VJ**
>
> We thank you for the comments, and we are happy to resolve any doubts.
>
> **W1: This paper combines the ideas of autoregressive graph diffusion [1] and block generation [2]. Although the combination is natural, I am not clear on the main difference between the proposed method and GRAN. It seems that the unity of autoregression and one-shot is due to the design of block generation, rather than the diffusion of node removal.**
>
> We are happy to clarify the difference with GRAN. Although the idea of using block generation is shared between IFH and GRAN, it has been adopted even earlier and falls into the category of motifs generation. Ours is a way to generalize the concept and unify it with sequential and one-shot generation to reap the benefits of both. On the other hand, the GRAN paper does not provide any information nor a way of integrating one-shot models into the sequentiality spectrum. The block generation mechanism is a way of controlling the degree of sequentiality, yes, but it is fixed and built into the model. Our approach allows the design of any Markov scheme for adding blocks, also with varying sizes, and has a direct link with the distribution in the dataset (as the insertion and halting models are trained accordingly). On the other hand, GRAN does not provide an explicit way to model the distribution of the number of nodes, fixing this number to the maximum size in the dataset, and delegating to the mixture of Bernoullis the task of generating connected components with some size. Furthermore, the GRAN approach might work for datasets with similar-sized graphs. Still, when there is a considerable variance, smaller graphs are penalized due to picking the biggest connected component of a generated graph, changing the output distribution of the model.
>
> Regarding your last point, you are partially correct, as the design of block generation and the diffusion of node removal are two faces of the same coin, which are the removal and insertion models. However, the block generation of GRAN is fixed and built into the model, and does not consider how the distribution of graph sizes of a dataset behaves.
>
> **W2: It's not clear to me what advantages 1-node IFH has over autoregressive methods. Does the benefit come from the prediction of the number of nodes?**
>
> Actually, the 1-node IFH is an autoregressive model, that is, any autoregressive model that generates one node at a time is a 1-node IFH (as pointed out in Section 4.1). In this case, the advantage is that, with our framework, we can adapt strong one-shot models for the 1-node generation modality, beating the autoregressive baselines, which usually implement tailored and simple models for inserting one node. We think this is one of the strongest capabilities of our IFH model.
>
> **W3: The time and memory costs of baselines are not reported in Tables 2 and 3. It is therefore impossible to see the trade-off between sampling quality and time.**
>
> Not quite, as our sampling quality/time trade-off analysis is intended for our framework, assessing what changes when we change the degree of sequentiality with the filler model in question. The comparison with the baselines in Table 2 aims to place our proposed approach in the state of the art, showing whether our framework is capable of generating performing models. To highlight this fact, we separated Table 2 into performance results and time/memory usage table.

---

### Official Review · Reviewer_9h17 · 2023-11-02

**Soundness:** 2 fair
**Presentation:** 3 good
**Contribution:** 2 fair
**Rating:** 3
**Confidence:** 3

**Summary:**

This paper explores combining autoregressive method with one-shot diffusion model. Diffusion model builds the forward process with adding noise gradually and the backward process with removing noise step-by-step. Similarly but not the same, the paper models the forward process as removing block of nodes and edges gradually towards an empty graph, and in backward process it reverts this process with adding nodes and edges back. This view combines autoregressive method and one-shot method together, via changing the granularity of node/edge removing. The author also discussed many different choice of node/edge removing random process. The experimental results show certain improvement on molecular datasets.

**Strengths:**

1. Exploring the direction of combining autoregressive method and one-shot diffusion model is meaningful, as they have different strength. The proposed method successfully combined them together, and the proposed process of block removing is interesting.
2. The author shows that the complexity of sequential model is lower than one-shot generation, and discussed its strength in section 4.3. This is interesting, and engineering wise one can use sparse storage for already generated components to save runtime and memory.
3. One key component of this proposed process is the block removing process, and the author discussed many choice with ablation studies.

**Weaknesses:**

1. The proposed method shares certain similarity with GRAN, while being novel for adapting diffusion process inside.
2. The goal of combining autoregressive method and one-shot generation is to combine their strength together while eliminate their shortcomings. However I think the proposed method is not ideal for this goal. For example, one-shot diffusion is a permutation equivariant generation model that is invariant to node permutation, here the designed model becomes ordering sensitive, which needs a careful ablation over node removing process. And autoregressive method has the problem of being hard to parallel during training, hence the designed model will be even slower in training comparing with one-shot generation. Last, the reported experimental result doesn't show a significant benefit of adapting sequential generation to one-shot diffusion.
3. The experimental result is kind of weak at current stage. First, for both QM9 and ZINC, the result doesn't beat the baseline like CDGS in many perspective. Second, for generic graph generation in Appendix, the designed method is significantly worse than the baseline. This questions whether the designed method, while being combination of autoregressive and one-shot, may suffers from the shortcoming of both sides instead of combining their strengths. Also, the designed method may suffer from the randomness of block removing process.
4. I suggest the author also discuss the training cost instead of just the test runtime and memory cost.

**Questions:**

1. For Table 2, there is no result for the baseline DiGress, is that equivalent to one-shot?
2. It seems that you have many different models trained: halting model, node size prediction model, and one denoising model. Can you talk about how do you do model selection for them?
3. You mentioned that you can use sparse format for already generated part, are you using this format during training?

---

> ### Author Response · Authors · 2023-11-22
> **Answer to reviewer 9h17 1/2**
>
> Thank you for your valuable comments, which we found interesting discussion points.
>
> **W1: The proposed method shares certain similarity with GRAN, while being novel for adapting diffusion process inside.**
>
> While it is true that our proposed method shares similarities with GRAN regarding block generation, our framework is capable of capturing every aspect of the data distribution, in particular the size of sampled graphs, thanks to learning a node-insertion and halting model. A notable limitation of GRAN is that it fixes the size of blocks as a hyperparameter, and the number of nodes generated to the maximum size of the dataset. This warps the output distribution, favoring bigger graphs, and making it challenging to generate smaller graphs. Additionally, IFH can be used to design any autoregressive block generation scheme, GRAN included.
>
> **W2: The goal of combining autoregressive method and one-shot generation is to combine their strength together while eliminate their shortcomings. However I think the proposed method is not ideal for this goal. For example, one-shot diffusion is a permutation equivariant generation model that is invariant to node permutation, here the designed model becomes ordering sensitive, which needs a careful ablation over node removing process. And autoregressive method has the problem of being hard to parallel during training, hence the designed model will be even slower in training comparing with one-shot generation. Last, the reported experimental result doesn't show a significant benefit of adapting sequential generation to one-shot diffusion.**
>
> While we agree that autoregressive models have the added burden of having to select the best node ordering, we found empirically that choosing the BFS order was enough to make our approach competitive with the best one-shot baselines.
>
> Regarding the fact that autoregressive models are hard to parallel during training: not entirely, as we sample the entire sequences from a batch of graphs, and train on the collected examples in parallel, because no hidden states are passed between steps. This characteristic was also implemented in GRAN, as it avoids the problem you mentioned. Following your suggestion, we added the time/memory tables for the training processes (Tables 2b, 2e, 3d) to show how sequentiality affects the time to reach convergence: our 1-node sequential model takes longer to train for a fixed amount of epochs, but reaches optimal validation performance much faster.
>
> On the last point, we showed in our experiments that adapting the one-shot model in question (DiGress) consistently improves performance for some degree of sequentiality (mostly 1-node sequentiality). One notable example is ZINC, where our 1-node sequential model surpasses CDGS on many metrics, and the original DiGress on all metrics.
>
> Motivated by these findings, we hypothesize that having a permutation equivariant model is not the only ingredient for a good-performing generative model. For example, breaking the sampling process into smaller steps might help reduce each atomic step's complexity. We have seen a similar phenomenon for diffusion models, sequentially denoising a sample in many steps, which beat fully one-shot models like VAEs and GANs.
>
> Our answer continues in the below comment.

---

> > ### Author Response · Authors · 2023-11-22
> > **Answer to reviewer 9h17 2/2**
> >
> > **W3: The experimental result is kind of weak at current stage. First, for both QM9 and ZINC, the result doesn't beat the baseline like CDGS in many perspective. Second, for generic graph generation in Appendix, the designed method is significantly worse than the baseline. This questions whether the designed method, while being combination of autoregressive and one-shot, may suffers from the shortcoming of both sides instead of combining their strengths. Also, the designed method may suffer from the randomness of block removing process.**
> >
> > Although we do not beat the strong one-shot baseline CDGS in every metric, there are many where we reach state-of-the-art results, mainly in the challenging ZINC dataset. Also, note that Uniqueness and Novelty are not so informative metrics at this stage (as all methods reach good results), and we included them to comply with earlier works.
> >
> > We wouldn't say that our method is significantly worse for generic graph generation, as it is still state-of-the-art in autoregressive models, surpassing GRAN, and competitive with CDGS.
> >
> > We think that our approach effectively combines the strengths of one-shot and autoregressive models, as we see a significant improvement over the latter category, and specifically over our one-shot variant. On the other hand, as you point out, some shortcomings are still present. For example, the great number of generation steps in 1-node sequential models can hinder performance and the effective learning of the halting model. We addressed this limitation in the new Discussion section 6. Another one can be the randomness of the block-removing process, which can be transformed into a strength if it is tailored for a specific task, e.g., motif generation in a molecular setup. This is entirely allowed by our framework.
> >
> > **W4: I suggest the author also discuss the training cost instead of just the test runtime and memory cost.**
> >
> > We thank you for the suggestion and added the "Training time/memory" tables (2b, 2e, 3d) on the side of each performance comparison table. Additionally, we discussed these tables in the "Time and memory consumption" paragraph of Section 5.1.
> >
> > **Q1: For Table 2, there is no result for the baseline DiGress, is that equivalent to one-shot?**
> >
> > We did not immediately include DiGress in our baselines because we are adapting the model into our framework. However, following your question, we added DiGress as one of the baselines for Table 2 for the following reason: in Section 5, we explain that the version incorporated comes without domain-specific knowledge and minor adjustments, so the one-shot variant is not quite DiGress. It is then meaningful to compare with it.
> >
> > **Q2: It seems that you have many different models trained: halting model, node size prediction model, and one denoising model. Can you talk about how do you do model selection for them?**
> >
> > We used the hyperparameters from the DiGress paper for the filler model, which showed good performance from the get-go. Regarding the halting and insertion model, we chose the best-performing set of hyperparameters from a random sample. Every selection choice was conducted on the validation set. Thanks to the modularity of our framework, each model can be trained and evaluated separately with respect to network hyperparameters.
> >
> > **Q3: You mentioned that you can use sparse format for already generated part, are you using this format during training?**
> >
> > Thanks for your interest in how we represent graphs. You are correct. We are using the sparse format also during training, in particular: (1) we build the whole removal sequence in sparse format, saving the removed subgraphs (and edges), (2) we train the halting and insertion models on the sparse representations; (3) we transform the removed subgraphs and edges to dense (applying the batching rules) and train the filler model. Motivated by your interest, we added in Appendix D.1 the two algorithms for training and generation of our framework, highlighting the usage of sparse and dense representations in the section.

---

### Official Review · Reviewer_HNVY · 2023-11-04

**Soundness:** 2 fair
**Presentation:** 2 fair
**Contribution:** 2 fair
**Rating:** 5
**Confidence:** 3

**Summary:**

This paper proposes a new Insert-Fill-Halt (IFH) framework for graph generation, which tries to bridge two types of existing approaches, i.e., one-shot generation and sequential generation. Specifically, at each step, the Insertion Model chooses how many new nodes to generate, the Filler Model fills the new nodes’ labels, features, and connections, and the Halt Model chooses if the generation needs to terminate. The training of the IFH framework uses the denoising diffusion model to develop a reversed node removal process, which destroys a given graph through many steps. Experimental results demonstrate the sample quality-time trade-off across a range of molecular and generic graphs datasets.

**Strengths:**

1. It is interesting to bridge one-shot and sequential graph generation methods with a unified framework.

2. Authors provide the analysis of the sample quality-time trade-off across many real-world datasets.

3. The paper is well-written and easy to understand.

**Weaknesses:**

1. The proposed framework does not provide insightful knowledge regarding choosing one-shot or sequential generation methods.

2. Only one base model is tested in the proposed IFH framework.

3. Experiments are not sufficient. Ablation studies are missing. The comparisons of time/memory cost with baselines are missing.

**Questions:**

Please see my listed weakness above.

---

> ### Author Response · Authors · 2023-11-22
> **Answer to reviewer HNVY**
>
> We thank you for the review and for acknowledging our work's main points. We are glad to resolve any doubt in the following.
>
> **W1: The proposed framework does not provide insightful knowledge regarding choosing one-shot or sequential generation methods.**
>
> We thank you for bringing up this point. We added a new "Discussion" section (numbered 6) to discuss further the empirical results, and how one might choose one-shot, sequential, or an in-between option.
> To summarize, our advice is to use one-shot models for smaller graphs, and go towards 1-node sequential models for bigger graphs to improve performance. When the graphs reach very large sizes, however, the halting signal becomes sparser, making the halting model harder to train. For these cases, going back to a hybrid solution can help, also in terms of memory and time consumption.
>
> Our proposed framework makes it possible to compare one-shot and sequential models that share the same architecture and generative model, with the degree of sequentiality becoming a hyperparameter. In fact, our experiments suggest that for different distributions of graphs, different levels of sequentiality will improve performance and resource usage. This is useful for the future design of graph generative models, as new architectures can be seamlessly adapted to sequential or one-shot generation, pushing their performances further than their native modality. Additionally, we discussed in Section 4.3 the complexity in time and memory of increasing sequentiality. Experimental results are consistent with the above statements.
>
>
> **W2: Only one base model is tested in the proposed IFH framework.**
>
> This is certainly a valid observation. One of our work's aims is to prove the feasibility of adapting a one-shot model to act in a sequential setup and observing whether this leads to an improvement. In the case of adapting DiGress, we got competitive results with the state-of-the-art one-shot models while beating by a significant margin autoregressive baselines, so we can say that the initial claim has had a positive outcome. We think adapting other one-shot models would be the next step, but we preferred to postpone it in favor of a more extended explanation of the framework.
>
> **W3: Experiments are not sufficient. Ablation studies are missing. The comparisons of time/memory cost with baselines are missing.**
>
> Regarding your first concern, we think we disguised the reader in dubbing the ablation study "Selection study". Actually, in that section we perform a similar study (on block generation formulation and different orders) as the one done in the GRAN paper under the section dubbed "Ablation study". If this is not resolving your issue, we would be happy to receive more information about what additional experiments we should perform.
>
> We gladly answer the second concern by referring to our selection study, which we clarified in Section 5.1, commenting on Table 1. There, we compared several instances of our model changing the removal process and node ordering in a grid. This is a selection study from the perspective of choosing the best combination of the two. At the same time, we are also isolating the contributions of the two factors, making it also an ablation study. In Section 5.1, we commented on such contributions, noting that both BFS and categorical removal independently improve performance, computational time, and memory. This latter analysis is characteristic of an ablation study. Furthermore, in the remainder of the experiments, we explore how the degree of sequentiality affects results.
>
> Regarding your last concern, in this work, we want to compare time/memory cost within a given model (DiGress in our case), varying the degree of sequentiality. With this analysis, we can actually understand the contribution of changing the degree of sequentiality.

---

### Author Response · Authors · 2023-11-22
**Answer to all reviewers**

We thank all reviewers for their insightful feedback and suggestions. We took in great consideration your comments, and revised our manuscript as follows:
1. We added a Discussion section (numbered 6) discussing the choice of the degree of sequentiality with respect to the dataset and the task. Limitations of our model are also presented.
2. We have cleared up the Experiments section, including our definition of degree of sequentiality. We also improved the explanation of our selection study and the discussion of the obtained results.
3. We expanded the experimental assessment by adding DiGress as a molecular baseline. Moreover, in order to give a more coherent presentation,  we have split Tables 2 and 3 into subtables for performance results (2a, 2d, 3b, 3c), subtables for training complexity (2b, 2e, 3d), and subtables for generation complexity (2c, 2f, 3e). We have also clarified in Table 3a what we mean by seq-1, seq-small, and seq-big for generic graphs generation.
4. We have moved the section devoted to generic graph generation into Appendix A, and expanded the discussion on our findings. We hope that this can make the section more accessible.
5. We added training and generation algorithms in Appendix D, together with a discussion on how DiGress has been adapted.
6. We slimmed down redundant parts of the paper (to make space for the new discussion section) and corrected typos.

We are currently running experiments on the GRAN Grid dataset. As soon as we get the results we will add them in the Appendix.

**Update:** due to technical difficulties we were not able to finalize the additional experiments on the Grid dataset. However, we believe that our method has already been evaluated on a significantly wide array of popular datasets, among which molecular and generic graphs. To the best of our knowledge, the Grid dataset has only been used in GRAN[1].

[1] Liao, Renjie, Yujia Li, Yang Song, Shenlong Wang, Will Hamilton, David K. Duvenaud, Raquel Urtasun, and Richard Zemel. "Efficient graph generation with graph recurrent attention networks." Advances in neural information processing systems 32 (2019).

---

### Meta-Review · Area_Chair_qxuc · 2023-12-05

**Metareview:**

This paper propose a d Insert-Fill-Halt (IFT) framework for graph generation, bridging the gap between one-shot graph generation with the sequential graph generation. Despite the interesting idea and import problem of graph generation it tackles, the manuscript may need another round of polish based on the reviewer's discussion for it to be published.

**Justification For Why Not Higher Score:**

The novelty is limited and the experiment was not done thoroughly to support the claims.

**Justification For Why Not Lower Score:**

N/A

---

### Decision · Program_Chairs · 2024-01-16

Reject